

# Towards a distributed connectionist account of cognates and interlingual homographs: evidence from semantic relatedness tasks

Eva D. Poort and Jennifer M. Rodd

Department of Experimental Psychology, University College London, London, United Kingdom

## ABSTRACT

**Background**. Current models of how bilinguals process cognates (e.g., "wolf", which has the same meaning in Dutch and English) and interlingual homographs (e.g., "angel", meaning "insect's sting" in Dutch) are based primarily on data from lexical decision tasks. A major drawback of such tasks is that it is difficult—if not impossible—to separate processes that occur during decision making (e.g., response competition) from processes that take place in the lexicon (e.g., lateral inhibition). Instead, we conducted two English semantic relatedness judgement experiments.

**Methods**. In Experiment 1, highly proficient Dutch–English bilinguals ($N = 29$) and English monolinguals ($N = 30$) judged the semantic relatedness of word pairs that included a cognate (e.g., "wolf"–"howl"; $n = 50$), an interlingual homograph (e.g., "angel"–"heaven"; $n = 50$) or an English control word (e.g., "carrot"–"vegetable"; $n = 50$). In Experiment 2, another group of highly proficient Dutch–English bilinguals ($N = 101$) read sentences in Dutch that contained one of those cognates, interlingual homographs or the Dutch translation of one of the English control words (e.g., "wortel" for "carrot") approximately 15 minutes prior to completing the English semantic relatedness task.

**Results**. In Experiment 1, there was an interlingual homograph inhibition effect of 39 ms only for the bilinguals, but no evidence for a cognate facilitation effect. Experiment 2 replicated these findings and also revealed that cross-lingual long-term priming had an opposite effect on the cognates and interlingual homographs: recent experience with a cognate in Dutch speeded processing of those items 15 minutes later in English but slowed processing of interlingual homographs. However, these priming effects were smaller than previously observed using a lexical decision task.

**Conclusion**. After comparing our results to studies in both the bilingual and monolingual domain, we argue that bilinguals appear to process cognates and interlingual homographs as monolinguals process polysemes and homonyms, respectively. In the monolingual domain, processing of such words is best modelled using distributed connectionist frameworks. We conclude that it is necessary to explore the viability of such a model for the bilingual case.

**Data, scripts, materials and pre-registrations.** . Experiment 1: http://www.osf.io/ndb7p; Experiment 2: http://www.osf.io/2at49.

Corresponding authors
Eva D. Poort, eva.poort.12@ucl.ac.uk
Jennifer M. Rodd, j.rodd@ucl.ac.uk

## INTRODUCTION

It is estimated that half of the world's population speaks more than one language, so one of the key issues in research on bilingualism is to determine how words are stored and accessed in a bilingual's mental lexicon. Much of the research so far seems to agree that all of the languages a bilingual or multilingual speaks are indeed stored in the same lexicon (this is known as the 'shared-lexicon' principle) and words from all of those languages become active during lexical retrieval (known as the 'non-selective access' principle) (for a review, see *Dijkstra, 2005*; *Dijkstra & Van Heuven, 2012*). Strong evidence in favour of these two claims comes from, among others, studies of cognate and interlingual homograph processing and research on cross-lingual priming of cognates and interlingual homographs. *Cognates* are words that exist in an identical (or near identical) form in two or more languages and refer to the same meaning, like the word "wolf" in Dutch and English. *Interlingual homographs* are words that, like cognates, share their form in more than one language, but that refer to a different meaning in those languages: "angel" means "insect's sting" in Dutch.

Briefly, these studies (see *Poort & Rodd, 2017b*, for a review) have shown that bilinguals process cognates more quickly than words that exist only in one of the languages they speak, such as the word "carrot" in English (which translates to "wortel" in Dutch). This effect is called the *cognate facilitation effect*. Interlingual homographs, in contrast, are often processed more slowly than single-language control words. Researchers use the term *interlingual homograph inhibition effect* to describe this finding. Furthermore, research has found that priming a cognate in one language tends to speed up processing of that cognate in the other language (*Bowers, Mimouni & Arguin, 2000*; *Cristoffanini, Kirsner & Milech, 1986*; *Gerard & Scarborough, 1989*; *Poort, Warren & Rodd, 2016*), while priming an interlingual homograph has been shown to slow down subsequent processing of that interlingual homograph in the other language (*Poort, Warren & Rodd, 2016*). Facilitation for cognates and inhibition for interlingual homographs would be unlikely to arise if the two language-specific readings of these words were stored in two separate lexicons. Similarly, cross-lingual priming would be unlikely to arise if the words in the two languages were stored separately. Therefore, both the existence of the cognate facilitation effect and interlingual homograph inhibition effect and of these cross-lingual priming effects provide evidence in favour of the idea that the mental lexicon in bilinguals contains words from both languages and that those languages are fully interconnected. Moreover, these phenomena raise the question of how cognates and interlingual homographs are represented in this shared and interconnected bilingual mental lexicon.

## The processing and representation of cognates and interlingual homographs in the bilingual lexicon

Most of the research to date has been concerned with the representation and processing of cognates. The cognate facilitation effect has most commonly been observed in visual lexical decision experiments (e.g., *Caramazza & Brones, 1979*; *Cristoffanini, Kirsner & Milech, 1986*; *De Groot & Nas, 1991*; *Dijkstra, Grainger & Van Heuven, 1999*; *Dijkstra et al., 2010*; *Duyck et al., 2007*; *Lemhöfer & Dijkstra, 2004*; *Peeters, Dijkstra & Grainger, 2013*; *Sánchez-Casas, García-Albea & Davis, 1992*; *Van Assche et al., 2009*; *Van Hell & Dijkstra, 2002*), though it has also been observed in word production (*Kroll et al., 1999*; *Costa, Caramazza & Sebastián-Gallés, 2000*; *Schwartz, Kroll & Diaz, 2007*), word association (*Van Hell & De Groot, 1998*; *Van Hell & Dijkstra, 2002*) and sentence reading (*Duyck et al., 2007*; *Libben & Titone, 2009*; *Schwartz & Kroll, 2006*; *Van Assche et al., 2009*; *Van Hell & De Groot, 2008*), although the effect is often smaller in the sentence reading experiments than in experiments that have presented the words in isolation.

In addition, as mentioned previously priming cognates in one of the languages a bilingual speaks often results in faster processing of those cognates in the other language. For example, both *Cristoffanini, Kirsner & Milech (1986)* and *Gerard & Scarborough (1989)* primed Spanish–English cognates (e.g., "reunion") by asking participants to first complete a task in either Spanish or English and then complete a second task (a lexical decision task) in the other language. Both teams of researchers found that participants responded more quickly to the cognates when they had been primed than when they had not. Similarly, *Bowers et. al*, (*2000*, Exp. 1) found that French–English bilingual participants recognised cognates in a French lexical decision task more quickly if they had studied these words in English earlier on during the experiment than if they had not. Finally, *Poort, Warren & Rodd (2016)* showed that a single encounter with a cognate in a Dutch sentence context facilitated recognition times of those cognates in an English lexical decision task approximately 16 min later for Dutch–English bilinguals.

Much research has also focused on the representation and processing of interlingual homographs. Like the cognate facilitation effect, this effect has been found most frequently in experiments investigating visual word recognition (De Bruijn et al., 2001; *Dijkstra, Van Jaarsveld & Ten Brinke, 1998*; *Kerkhofs et al., 2006*; *Lemhöfer & Dijkstra, 2004*; *Van Heuven et al., 2008*), but it has also been observed in auditory word recognition (using interlingual homophones; *Lagrou, Hartsuiker & Duyck, 2011*; *Schulpen et al., 2003*), word production (*Jared & Szucs, 2002*; *Smits et al., 2006*) and sentence reading (*Libben & Titone, 2009*; *Titone et al., 2011*).

Priming studies of interlingual homographs have generated mixed results. For example, *Lalor & Kirsner (2001)* found no effects of cross-lingual priming on interlingual homographs, while *Gerard & Scarborough (1989)* report facilitative priming for their interlingual homographs. However, it appears that priming was facilitative when the priming language had been the participants' second language (English) and the testing language their first (Spanish). When the interlingual homographs were primed in Spanish and tested in English, priming appeared to be disruptive. In line with these findings, *Poort, Warren & Rodd (2016)* found that their Dutch–English bilingual participants responded

more slowly in their English lexical decision task to interlingual homographs that they had encountered before in a Dutch sentence than to those they had not seen before.

The model that is cited most often to account for the cognate facilitation effect and the interlingual homograph inhibition effect is the Bilingual Interactive Activation plus (BIA+) model (*Dijkstra & Van Heuven, 2002*), which was recently updated to include an account of bilingual word production as well as bilingual word recognition and renamed the Multilink model (*Dijkstra et al., 2018*). According to the BIA+ and Multilink models, the bilingual word recognition system consists of two components: a word identification system (which was inspired by *McClelland & Rumelhart*'s (*1981*) and *Rumelhart & McClelland*'s (*1982*) Interactive Activation model) and a task system (inspired by *Green*'s (*1998*) Inhibitory Control model). In both models, in the word identification system, words consist of an orthographic node, a phonological node and a semantic node. For the purposes of our research, we will focus only on the orthographic and semantic representations for now. Furthermore, it is important to note that neither model currently has considered how long-term cross-lingual priming effects might emerge.

In both the BIA+ and Multilink model, words also have a resting activation level that depends on their frequency of use and, depending on their similarity to the input word, become more (or less) activated until one is selected. In the BIA+ (but not the Multilink) model, the orthographic nodes of words inhibit each other through a process called lateral inhibition, irrespective of the language to which they belong. After a period of competition, a word is recognised when it passes the recognition threshold. In the current Multilink model, the lateral inhibition parameter has been set to 0, so words do not compete with each other and word recognition simply occurs when a word reaches the recognition threshold. The task system in both models continuously reads out the activation in the word identification system and weighs the different levels of activation to arrive at a response relevant to the task at hand.

According to *Peeters, Dijkstra & Grainger (2013)*, in the BIA+ model, cognates consist of a shared orthographic representation and a shared semantic representation. Two language-specific morphemes or tags store information about the cognates' frequency and morpho-syntactic characteristics in each language separately. In the Multilink model, the representation of cognates is slightly different, as they consist of two identical orthographic nodes and a shared semantic node. Nevertheless, both models assume that the cognate facilitation effect is a consequence of greater activation at the semantic level. In the BIA+ model this is due to resonance between the shared semantic node and the shared orthographic node (*Dijkstra et al., 2010*; *Peeters, Dijkstra & Grainger, 2013*); in the Multilink model this is because the semantic node receives 'extra' activation from the two identical orthographic nodes (*Dijkstra et al., 2018*).

In contrast, in the BIA+ model, the two readings of an identical interlingual homograph share none of their representations: both readings have their own orthographic and semantic nodes (*Dijkstra & Van Heuven, 2002*; *Kerkhofs et al., 2006*). The interlingual homograph inhibition effect is explained by this model as a disproportionately strong effect of lateral inhibition: as with any two words, the two orthographic nodes of the interlingual homograph laterally inhibit each other. However, because these representations

are identical, this competition is stronger than that between two regular words. Because the Multilink model no longer includes a lateral inhibition mechanism, it does not currently offer an account of interlingual homographs (*Dijkstra et al., 2018*).

## The limitations of current models of the bilingual mental lexicon

The models discussed above are based primarily on data from single-language lexical decision tasks. In a single-language lexical decision task, participants have to decide whether letter strings are words in a *specific* language (usually the bilingual's second language). Research has shown that the size of the cognate facilitation effect and the interlingual homograph inhibition effect in such tasks depends on the other types of stimuli included in it. For example, a processing cost for interlingual homographs compared to control words is more likely to be observed when the experiment also includes words from the bilingual's other language (i.e., the non-target language, usually the bilingual's first language) that require a "no"-response *De Groot, Delmaar & Lupker, 2000*; *Dijkstra et al., 2000*; *Dijkstra, Van Jaarsveld & Ten Brinke, 1998*; *Poort & Rodd, 2017b*; *Von Studnitz & Green, 2002*). In single-language lexical decision tasks that do not include words from the non-target language, interlingual homographs are often processed as quickly as control words. It is thought that the inhibition effect observed in such tasks is greater because of response competition between the English reading of the interlingual homograph, which is linked to the "yes"-response, while the Dutch reading is linked to the "no"-response.

Similarly, *Poort & Rodd (2017b)* and *Vanlangendonck (2012)* have demonstrated that the cognate facilitation effect in lexical decision tasks is strongly reduced when that task includes non-target language control words. In *Poort & Rodd*'s (*2017b*) experiments, for example, Dutch–English bilinguals completed one of five versions of an English lexical decision task. The standard version of the task included a set of cognates, English controls and 'regular' non-words. Three of the other versions also included either a set of interlingual homographs, a set of pseudohomophones (instead of the regular non-words) or a set of Dutch words that participants were required to respond "no" to. The fifth version included all three of these extra stimuli types. In any versions of the experiment that included the Dutch words, there was no cognate facilitation effect. Furthermore, in an exploratory analysis, a large inhibition effect was found for cognates that were immediately preceded by a Dutch word. *Poort & Rodd (2017b)* concluded that response competition between the two readings of the cognates most likely reduced the facilitation effect in these versions, as it increased the interlingual homograph inhibition effect in the experiments discussed above. Crucially, this means it is difficult, if not impossible, to determine whether the facilitation and inhibition effects traditionally observed for cognates and interlingual homographs in lexical decision tasks originate in the lexicon or whether they are decision artefacts.

In particular, *Poort & Rodd*'s (*2017b*) experiments showed that the specificity of the representation of a word that is retrieved during lexical decision depends on the other types of stimuli included in the experiment. When the lexical decision task in those experiments did not include non-target language words, participants were able to respond based on a general sense of familiarity or 'word-likeness' of the stimuli. When the lexical decision task did include such words, however, they had to decide whether a given stimulus

was a word in a particular language or not. In these tasks, participants could only respond accurately if they accessed a specific representation of the word in their mental lexicon, which included information about the word's (written and spoken) form, its meaning and its language membership.

Furthermore, *Poort & Rodd* (*2017a*) note that the use of a lexical decision task may explain why they did not replicate *Poort, Warren & Rodd*'s (*2016*) long-term cross-lingual priming effects for cognates or interlingual homographs. They argue, based on evidence collected by Rodd and colleagues (*Rodd et al., 2013*; *Rodd et al., 2016*), that the long-term priming effect *Poort, Warren & Rodd (2016)* observed may primarily rely on access to semantics, which is not always required in a lexical decision task. Indeed, the use of a lexical decision task may also explain why the evidence for a cross-lingual priming effect for interlingual homographs is mixed. Access to semantics is not always required in a lexical decision task, but this may be crucial to observing a cross-lingual priming effect for interlingual homographs, as they have different meanings in the languages priming language than in the testing language.

In sum, in a lexical decision task, participants can use different sources of information or access different levels of representation to decide whether stimuli are words or not, depending on the level of specificity required by the task. In addition, some of these sources of information (like information about a word's form or meaning) will exert facilitatory or inhibitory influences within the lexicon, while others (like information about a word's language membership) will play a role in decision processes. With lexical decision tasks, it is challenging to separate these decision-level processes from lexicon-based processes and it is not always clear which level(s) of representation a particular task is tapping in to. Necessarily, then, lexical decision tasks should be used with caution and in combination with other tasks to obtain converging evidence regarding how people process language. To develop a unified theory of how bilinguals process cognates and interlingual homographs and how these words are stored in the bilingual mental lexicon, we need to use many different types of tasks that tap into different levels of representation, to allow us to unpick lexicon-based processes and task-based processes.

## The advantages of semantic relatedness tasks

To this end, the experiments we present here use a semantic relatedness judgement task to study the processing of cognates and interlingual homographs. During such a task, participants see pairs of words and are asked to decide as quickly and accurately as possible whether the words in each pair are related to each other in meaning or not. For example, participants may see the word "goat" followed by the word "sheep" (for a related response) or the word "wardrobe" (for an unrelated response). Evidently, this type of task requires access to the semantic representation of both the initial word (the target) and the subsequent word (the probe). Furthermore, although participants respond to the probe, the participants' performance on this task also reflects how quickly and easily the target is processed and which meaning(s) is/are initially accessed, as long as the probe is presented before processing of the target has finished.

Aside from requiring access to semantics, another advantage of using a semantic relatedness task is that, in such a task, participants are more likely to use the same sources of information regardless of the demands of the task or the other stimuli included in it. To decide whether two words are related to each other in meaning, a participant must always access a representation of both the forms of these words and, through these forms, the meanings of those words. The characteristics of the other stimuli included in the task may make the decision of whether those two words are related or not more difficult (e.g., if some pairs are less obviously related than others), but those characteristics do not influence whether only the form representation or also the meaning representation is accessed, as can be the case in a lexical decision task. In this way, a semantic relatedness task is also more similar to natural language processing than a lexical decision task. In particular, although the task may still be artificial in nature, the end point of lexical access is the same: information about a word's form is used in order to access the relevant information about its meaning. Therefore, a semantic relatedness task is more likely to be a truer reflection of the processes that occur in the mental lexicon during natural language processing than a lexical decision task.

Furthermore, decisions in a semantic relatedness task are less likely to be based on language membership information, the main driving factor for the emergence of response competition in lexical decision tasks. In single-language lexical decision tasks that include non-target language words, language membership information is inherent to the decision that participants are required to make. However, even in single-language lexical decision tasks that do not include non-target language words, participants may rely extensively on language membership information, as they have been instructed to decide if the stimulus is a word in a particular language. In a semantic relatedness task, it is less clear what role language membership plays. In particular, when participants see an interlingual homograph (e.g., "angel") and an English probe (e.g., "heaven"), they may use the fact that the task is in English to allow them to ignore information about the interlingual homograph's Dutch meaning (i.e., "insect's sting"). However, they may also base their decisions purely on the relationship between the meaning of the target and the probe and this is arguably more likely. In other words, response competition in a semantic relatedness task is more likely to reflect lexico-semantic competition: the fact that the concept of "insect's sting" is not related to the concept of "heaven" would warrant a "no"-response, which would compete with the "yes"-response that is required because the concept of "spiritual being" is related to "heaven".

Despite these advantages, it seems that semantic relatedness tasks have not been commonly used in bilingual research. Only two studies appear to have used such a task to examine cognate processing and their results are contradictory. *Degani, Prior & Hajajra (2018)* asked a group of Arabic–Hebrew bilinguals and native Hebrew speakers to complete a semantic relatedness task in Hebrew that included, among others, pairs of related words of which the first word was either a cognate or a control word. Note that Arabic and Hebrew use a different script, so the cognates in this experiment were pronounced very similarly or even the same but were not written the same. Nevertheless, they found that the Arabic–Hebrew bilinguals responded more accurately and marginally

more quickly on the cognate trials compared to the control trials, while the native Hebrew speakers did not.

In contrast, *Yudes, Macizo & Bajo (2010)* recorded EEG signals while Spanish–English bilinguals completed a Spanish semantic relatedness task and found that their participants did not respond significantly more quickly or slowly to pairs that included a cognate than to pairs that included a control word. Furthermore, there were no differences in the N400 component between the cognate trials and the control trials, which is thought to reflect the process of semantic integration and lexical factors such as word frequency (*Yudes, Macizo & Bajo, 2010*). However, *Yudes, Macizo & Bajo (2010)* included mainly non-identical cognates in this experiment: only eight of the 100 cognates were identically spelled in Spanish and English. Research has shown that the cognate facilitation effect is greater for identical cognates than for non-identical cognates (*Duyck et al., 2007*; *Dijkstra et al., 2010*; *Van Assche et al., 2011*; *Comesaña et al., 2015*). Furthermore, the effect is often greater in the bilingual's second language than in their first language (*Van Hell & Dijkstra, 2002*). In other words, *Yudes, Macizo & Bajo (2010)* may not have found evidence for a cognate facilitation effect because they used almost exclusively non-identical cognates and tested in the participants' native language.

It also appears that only three studies have used a semantic relatedness task to examine interlingual homograph processing. In the same experiment discussed above, *Degani, Prior & Hajajra (2018)* also included pairs of words which included interlingual homophones. The probe for these interlingual homophones was always related to the *non-target language* (Arabic) meaning of the interlingual homophone and so unrelated its meaning in the *target language* (Hebrew). They found that the Arabic–Hebrew bilinguals but not the native Hebrew speakers responded less accurately on the interlingual homophone trials than on the control trials. Furthermore, in a cross-modal version of this task, *Prior et al. (2017)* found that their Arabic–Hebrew participants also responded more slowly to pairs that included an interlingual homophone, while their native Hebrew speakers did not.

Similarly, *Macizo, Bajo & Cruz Martín (2010)* asked Spanish–English bilinguals to make semantic relatedness judgements in *English* to identical interlingual homographs paired with probes that were related to the *Spanish* meanings of the interlingual homographs and unrelated to the English meanings (e.g., "pie"–"toe", where "pie" means "foot" in Spanish). Their participants also responded less accurately and more slowly on interlingual homograph trials than on control trials (e.g., "log"–"toe"). These studies suggest that the participants accessed the non-target language meanings of the interlingual homophones/-graphs and then inhibited this. They indicate that the disadvantage for interlingual homographs in lexical decision is not solely an artefact of using a lexical decision task. However, all of the interlingual homographs were paired with probes that were related to the non-target language meaning but unrelated to the target-language meaning, which likely made the participants rely more on language membership information. Response competition based on language membership information may, therefore, have played a role in this effect.

A final advantage of using a semantic relatedness task is that, in the monolingual domain, this task has also been used successfully to replicate effects of long-term word-meaning priming, the monolingual equivalent of *Poort, Warren & Rodd*'s (*2016*) long-term cross-lingual priming paradigm. Initial experiments using this paradigm used a word association task to show that a single encounter with an ambiguous word's less-used (subordinate) meaning can bias future interpretation of that word towards that meaning (*Betts et al., 2017*; *Rodd et al., 2013*; *Rodd et al., 2016*). For example, *Rodd et al. (2013)* found that participants who had encountered the sentence "The man accepted the **post** in the accountancy firm" were more likely to provide words related to the "job"-meaning of "post" than the "mail"-meaning than if they had encountered the sentence "The man accepted the **job** in the accountancy firm". *Gilbert et al. (2018)* used a semantic relatedness task to replicate these findings: participants who had encountered the sentence "The pig **pen** was muddier than ever" responded more quickly to the probe "enclosure" for "pen" than participants who had not. Therefore, if cross-lingual priming is driven by similar mechanisms as within-language priming, we would expect this type of task to be sensitive to cross-lingual priming manipulations as well.

## The present experiments

To summarise, there are reasons to think that lexical decision tasks are problematic to study language processing, including how cognates and interlingual homographs are processed and represented in the bilingual mental lexicon. Semantic relatedness tasks have the advantage of requiring access to meaning representations, while lexical decision tasks do not always involve semantic processing of the stimuli. The specificity of the representation that is accessed during a semantic relatedness task is also less likely to depend on task demands and stimulus list composition and the decision is less likely to be based on language membership information. Finally, a semantic relatedness task has been used successfully to show evidence of long-term priming in the monolingual domain (*Gilbert et al., 2018*). Therefore, and because this type of task has not been used often in bilingual research in the past, we decided to use a semantic relatedness judgement task to study cognate and interlingual homograph processing.

In the two experiments presented here, participants were asked to judge whether a target word—either a cognate (e.g., "wolf"), interlingual homograph ("angel") or English control ("carrot")—was semantically related to a subsequent probe. In Experiment 2, the participants also first completed a task in Dutch in which they read sentences that contained some of these cognates, interlingual homographs or the Dutch translation of an English control. All targets were paired with related probes, because a "yes"-response in that case clearly signals that the participant accessed the relevant (target-language) meaning of both the target and the probe. In addition, none of the interlingual homographs were paired with probes that were related only to their non-target language (Dutch) meaning, to avoid participants automatically linking the non-target language meanings of the interlingual homographs to the "no"-response and, thus, eliciting language membership-based response competition. As all targets were paired with related probes, additional filler items were included that were paired with unrelated probes. The target was presented before the probe

to prevent biasing the interpretation of the target towards the meaning of the probe, which could have negated any effects. The target was presented only briefly (200 ms) and almost immediately followed by the probe (50 ms after target offset) so that processing of the target would not be finished before the probe was presented. Consequently, the task would still reflect processing of the targets even though participants were responding to the probes.

## EXPERIMENT 1

Experiment 1 was pre-registered as part of the Center for Open Science's Preregistration Challenge (http://www.cos.io/prereg/). The stimuli, data and processing and analysis scripts can be found on the Open Science Framework (http://www.osf.io/ndb7p/), as well as an Excel document with detailed results from all of the analyses. The preregistration can be retrieved from http://www.osf.io/u2fyk/ (*Poort & Rodd, 2017c*). Where applicable, deviations from the pre-registration will be noted. Also note that, unless specified, for all analyses reported in this article the significance level was set at .05.

### Introduction

The aim of Experiment 1 was to examine semantic processing of cognates and interlingual homographs and to determine whether the cognate facilitation effect and interlingual homograph inhibition effect observed in lexical decision are a result of how these words are stored in the bilingual lexicon or whether they are merely (lexical decision) task artefacts. A group of native monolingual British English speakers performed the same experiment as a group of Dutch–English bilinguals to rule out the possibility that the effects seen in the bilingual group were due to pre-existing differences between the three word types. To further minimise the role that language membership information would play in the semantic relatedness task (for example by prompting the bilingual participants to adopt a more bilingual language processing mode; (*Grosjean, 1998*; *Soares & Grosjean, 1984*), Experiment 1 was advertised and conducted entirely in the language of the task (English), so that the Dutch–English bilinguals would not assume that their knowledge of the stimuli in Dutch would be relevant to the task.

Our predictions were based on the assumption that the cognate facilitation effect and interlingual homograph inhibition effect are not merely (lexical decision) task artefacts but instead point to a special status of cognates and interlingual homographs in the bilingual lexicon, as outlined by the BIA+ model (*Dijkstra & Van Heuven, 2002*) and the Multilink model (*Dijkstra et al., 2018*). We predicted that when our bilingual participants encountered a cognate in this task, they would quickly and easily access its shared meaning representation. When they then saw the probe, because the cognate's meaning would still be strongly activated (more strongly than if they had seen an English control) they would also be able to relatively quickly and easily decide that the probe's meaning was related to the cognate's meaning. In contrast, we predicted that when our participants encountered an interlingual homograph, initially they would access both its Dutch and English meaning representation. When the probe was then presented, both meanings would still be activated and lexico-semantic competition between these meanings would slow the participants down and make them more prone to mistakes, resulting in an interlingual homograph inhibition

effect. We predicted no differences between the cognates, interlingual homographs and English controls for the monolingual participants.

## Methods

### Participants

Our aim was to recruit at least 30 participants for each group. The bilingual participants were recruited first and had to be between the ages of 18 and 35, of Dutch or Belgian nationality and resident in the Netherlands or Belgium at the time of the experiment. Their native language had to be Dutch and they had to be fluent speakers of English, with no diagnosis of any language disorders. Prolific Academic (*Damer & Bradley, 2014*; http://www.prolific.ac) was used for participant recruitment. Only participants who had previously indicated on Prolific that these criteria applied to them were able to access the experiment. A total of 31 bilingual participants who met these criteria was recruited. (One other participant in hindsight did not meet all of these criteria, based on the demographics questions included in the experiment.)

The monolingual participants were recruited next and efforts were made to match the two groups in terms of age and educational profile. The monolingual participants had to be native speakers of British English who spoke no other languages fluently and were not diagnosed with any language disorders. They had to be between the ages of 18 and 31 (the age range of the bilingual participants), hold the British nationality and be resident in the United Kingdom at the time of the experiment. Again, only participants who had indicated previously on Prolific that these criteria applied to them were able to access the experiment. To match on education, the bilingual participants were classified as having obtained a 'high' educational degree if they had completed or were currently enrolled in an undergraduate or higher degree or as having obtained a 'low' educational degree if they had not. The monolingual participants were recruited in roughly equal proportions of high and low educational degrees as the bilinguals, based on the degree of education the participants had indicated on Prolific they had completed. A total of 31 monolingual participants was recruited. (Three additional participants had been recruited who in hindsight did not meet the eligibility criteria, as indicated by their answers to the demographics questions.) All participants gave informed consent (by means of ticking a box in the online consent form) and were paid £3 for their participation in the experiment. The UCL Experiment Psychology Ethics Committee provided approval of our study protocol (Project ID: EP/2017/009).

There were two stages at which participants could be excluded from our analyses. First, while testing was still on-going, participants who scored less than 80% correct on the semantic relatedness task and/or less than 50% on either or both of the two language proficiency measures (the Dutch and English LexTALEs; *Lemhöfer & Broersma, 2012*) were excluded and replaced. Two participants (one in each group) were excluded at this stage and two new participants tested in their stead. Second, after testing had finished and a total of 60 useable datasets had been gathered, each participant's performance on the targets (cognates, interlingual homographs and English controls) included in the semantic relatedness task was compared to the mean of their group to determine whether any participants were outliers with respect to their group's overall performance. One bilingual

participant had performed worse than three standard deviations below the bilingual group's mean (68.0%, $M = 89.2\%$, $SD = 6.9\%$) and was excluded at this stage.

The remaining 29 bilingual participants (18 males; $M_{age} = 22.4$ years, $SD_{age} = 3.8$ years) had started learning English from an average age of 7.4 ($SD = 2.3$ years) and so had an average of 15.0 years of experience with English ($SD = 4.2$ years). They rated their proficiency as 9.3 out of 10 in Dutch ($SD = 0.9$) and as 8.5 in English ($SD = 0.7$). A two-sided paired $t$-test showed this difference to be significant [$t(28) = 5.010$, $p < .001$]. These self-ratings were confirmed by their high LexTALE scores in both languages, which a two-sided paired $t$-test showed were also higher in Dutch [Dutch: $M = 87.2\%$, $SD = 6.3\%$; English: $M = 82.5\%$, $SD = 9.5\%$; $t(28) = 2.406$, $p = .023$].

The 30 included monolingual participants (10 males; $M_{age} = 26.2$ years, $SD_{age} = 3.7$ years) had scored 8.4 percentage points higher on the English LexTALE than the bilingual participants. A two-sided independent-samples Welch's $t$-test showed this difference to be significant [bilinguals: $M = 82.5\%$, $SD = 9.5\%$; monolinguals: $M = 91.0\%$, $SD = 6.1\%$; $t(47.7) = -4.029$, $p < .001$]. Twenty-three (out of 29) bilingual and twenty-five (out of 30) monolingual participants had obtained high education degrees according to our classification. A chi-square test showed this difference to be non-significant [$\chi^2(1) = 0.152$, $p = .697$; $\alpha = .01$]. In terms of age, there was a small but significant difference of 3.8 years between the two groups, with the monolingual participants being older than the bilingual participants [$t(56.7) = -3.891$, $p < .001$; $\alpha = .01$]. As per our preregistration, all analyses that involved group as a factor were conducted both with age included as a covariate and without.

### Materials

The full set of stimuli can be found in the stimuli.xlsx document in the Experiment materials component of our OSF project (http://www.osf.io/6nehr).

*Targets & probes* The same set of 50 identical cognates, 50 identical interlingual homographs and 50 English controls included in Experiment 2 of *Poort & Rodd*'s *(2017a)* study was used in this experiment. (Note that *Poort & Rodd (2017a)* also included a set of 50 non-identical cognates which were not included in this experiment.) These items comprised a subset of the 56 cognates, 56 interlingual homographs and 56 English controls that *Poort & Rodd (2017b)* had also used. The items ranged in frequency from 0.98 and 590.69 occurrences per million (according to the SUBTLEX-US database; *Brysbaert & New, 2009*), were between 3 and 8 letters long and had OLD20 values (*Yarkoni, Balota & Yap, 2008*) between 1 and 2.80. *Poort & Rodd (2017b)* had also pre-tested these items in terms of their meaning, spelling and pronunciation similarity in Dutch and English across two pre-tests in which a total of 90 Dutch–English bilinguals participated, to ensure that the items belonged to their intended word type categories.

The three word types had been previously matched on English log-transformed frequency (weight: 1.5), word length (weight: 1.0) and OLD20 (weight: 0.5) using the software package Match (*Van Casteren & Davis, 2007*)[1] . Independent-samples two-tailed Welch's $t$-tests indeed showed that the differences between the three word types on the matching criteria

[1] *Poort & Rodd (2017a)* weighted the matching variables in this order as it has been shown that frequency is a more important predictor of lexical decision reaction times than word length and orthographic complexity (*Brysbaert et al., 2016*; *Keuleers et al., 2015*; *Yarkoni, Balota & Yap, 2008*).

were not significant (all $ps > .05$). An analysis of the meaning similarity ratings confirmed that the cognates differed significantly from the interlingual homographs ($p < .001$), but not from the English controls ($p = .737$). The cognates and interlingual homographs were significantly different from the English controls in terms of spelling similarity ratings (both $ps < .001$) but not each other ($p = .109$). All three word types differed significantly from each other in terms of pronunciation similarity ratings (all $ps < .03$). Table 1 lists means and standard deviations per word type for each of the matching criteria (and word frequency in occurrences per million) for both English and Dutch (for the sake of completeness), as well as the meaning, spelling and pronunciation similarity ratings obtained by *Poort & Rodd (2017b)*.

Each of the 150 target items was assigned a related probe (e.g., "wolf"–"howl") for the semantic relatedness task. The probes were of roughly equal frequency, length and orthographic complexity as the targets themselves. They ranged in frequency from 0.98 to 866.04 occurrences per million, were between 3 and 9 letters long and had OLD20 values between 1 and 3.75. Means and standard deviations of these variables for each set of probes per word type can be found in Table 2. The sets of probes for the three word types did not significantly differ from each other in terms of log-transformed frequency, word length or OLD20 (all $ps > .2$).

*Fillers*  As all of the targets were assigned related probes, the experiment also included 150 unrelated filler–probe pairs. These pairs consisted of 105 filler words pseudo-randomly selected using random word generator websites and paired with another 105 pseudo-randomly generated unrelated probes (e.g., "peanut"–"jazz"), as well as an additional 15 cognate fillers, 15 interlingual homograph fillers and 15 English control fillers also paired with unrelated probes (e.g., "mug"–"margin"). The additional cognates, interlingual homographs and English controls had been selected from the remaining materials that *Poort & Rodd (2017b)* had pre-tested and were paired with unrelated probes to ensure that the participants would not assume that any cognate or interlingual homograph always required a "yes"-response.

*Pilot experiment*  To verify that each item was indeed related or unrelated to its chosen probe as intended, a pilot of the semantic relatedness task was conducted with a group of 16 monolinguals who did not take part in the main experiment. Two participants were excluded, as they had scored less than 80% correct on the task. The data from the remaining 14 participants (2 male; $M_{age} = 33.1$ years, $SD_{age} = 8.8$ years) indicated that overall accuracy for the related trials was high ($M = 93.4\%$; $SD = 9.1\%$), as was performance on the unrelated trials ($M = 96.7\%$, $SD = 7.3\%$). The probes for three items (two targets and one filler) with a percentage correct of less than 70% were changed. The description of the stimuli above refers to the items as they were used in the experiment, including any changes made after the pilot.

### Design and procedure

The experiment employed a mixed design. Word type was a within-participants/between-items factor: participants saw all words of each word type, but each word of course belonged

**Table 1  Experiment 1 & 2. Means (*SD*s) of the Dutch and English characteristics, similarity ratings and sentence length for the cognates, interlingual homographs, English controls and fillers.**  Frequency refers to the SUBTLEX word frequency in occurrences per million [see *Keuleers, Brysbaert & New (2010)* for Dutch and *Brysbaert & New (2009)* for English]; log10(frequency) refers to the SUBTLEX log-transformed word frequency [log10(raw frequency + 1)]; OLD20 refers to *Yarkoni, Balota & Yap*'s (*2008*) measure of orthographic complexity of a word, expressed as its mean orthographic Levenshtein distance to its 20 closest neighbours. The similarity ratings were obtained by *Poort & Rodd (2017b)* and were given on a scale from 1 (not at all similar) to 7 [(almost) identical]. Sentence length refers to the length of the Dutch prime sentences shown during the Dutch semantic relatedness task of Experiment 2. The Dutch characteristics of the cognates, interlingual homographs and Dutch translations of the English controls were relevant for Experiment 2 only, as was prime sentence length.

| | Dutch characteristics | | | | English characteristics | | | | Similarity ratings | | | |
|---|---|---|---|---|---|---|---|---|---|---|---|---|
| | frequency | log10 (frequency) | word length | OLD20 | frequency | log10 (frequency) | word length | OLD20 | meaning | spelling | pronunciation | sentence length |
| cognates | 39.0 (59.4) | 2.92 (0.49) | 4.48 (0.97) | 1.58 (0.40) | 44.8 (57.0) | 3.10 (0.48) | 4.48 (0.97) | 1.59 (0.31) | 6.85 (0.22) | 7.00 (0.00) | 5.87 (0.68) | 9.46 (1.64) |
| interlingual homographs | 39.2 (82.9) | 2.70 (0.68) | 4.30 (0.93) | 1.34 (0.35) | 56.2 (108) | 2.98 (0.66) | 4.30 (0.93) | 1.46 (0.32) | 1.15 (0.28) | 7.00 (0.01) | 5.53 (0.79) | 9.20 (1.81) |
| English controls | 27.8 (26.3) | 2.88 (0.46) | 4.68 (0.87) | 1.43 (0.31) | 30.2 (26.0) | 3.02 (0.41) | 4.46 (0.93) | 1.57 (0.31) | 6.86 (0.19) | 1.12 (0.24) | 1.11 (0.21) | 9.32 (1.30) |
| fillers | — | — | — | — | 54.3 (142) | 2.97 (0.58) | 4.42 (1.07) | 1.58 (0.36) | — | — | — | — |

**Notes.**

The English controls were called translation equivalents in Experiment 2.

**Table 2 Experiment 1 & 2. Means (SDs) of the English characteristics of the probes in the semantic relatedness task for the cognates, interlingual homographs, English controls and fillers.** Frequency refers to the SUBTLEX word frequency in occurrences per million (*Brysbaert & New, 2009*); log10(frequency) refers to the SUBTLEX log-transformed raw word frequency [log10(raw frequency=1)]; OLD20 refers to *Yarkoni, Balota & Yap*'s (*2008*)'s measure of orthographic complexity of a word, expressed as its mean orthographic Levenshtein distance to its 20 closest neighbours.

| | frequency | log10(frequency) | word length | OLD20 |
|---|---|---|---|---|
| cognates | 66.3 (113) | 3.04 (0.69) | 5.34 (1.39) | 1.92 (0.62) |
| interlingual homographs | 60.7 (117) | 2.98 (0.67) | 5.14 (1.37) | 1.80 (0.46) |
| English controls | 55.1 (109) | 2.98 (0.64) | 5.34 (1.53) | 1.90 (0.61) |
| fillers | 50.1 (113) | 2.89 (0.63) | 5.11 (1.23) | 1.85 (0.49) |

**Notes.**

OLD20 information was not available for the probe "logo" for the interlingual homograph "brand"–"brand". The English controls were called translation equivalents in Experiment 2.

to only one word type. Group was a between-participants/within-items factor: participants belonged either to the bilingual or the monolingual group, but each item was seen by both groups.

The experiment was created and conducted using the Gorilla Experiment Builder (*Anwyl-Irvine et al., 2019*; http://www.gorilla.sc). The experiment protocol and all tasks are available to preview and copy from Gorilla Open Materials at http://www.gorilla.sc/openmaterials/36778. It comprised two (for the monolinguals) or three (for the bilinguals) separate tasks: (1) the English semantic relatedness task, (2) the English version of the LexTALE (*Lemhöfer & Broersma, 2012*) and, for the bilingual participants only, (3) the Dutch version of the LexTALE (*Lemhöfer & Broersma, 2012*). As mentioned previously, the bilingual participants were not told that they were being recruited because of their status as native Dutch speakers or that Dutch would be in any way relevant to the experiment, until after they had completed the semantic relatedness task. To achieve this, the consent form at the start of the experiment was in English, as was the study description on Prolific. At the end of the experiment, the participants completed a self-report language background survey (in Dutch for the bilinguals or English for the monolinguals). The answers to these questions were used to verify eligibility (see the Participants section).

The semantic relatedness task was identical for the bilingual and monolingual participants. During the semantic relatedness task, the participants saw all 150 related target–probe pairs ("yes"-responses) and all 150 unrelated filler–probe pairs ("no"-responses) and were asked to indicate, by means of a button press, as quickly and accurately as possible, whether the word they saw first was related in meaning to the word they saw second. A practice block of 6 pairs was followed by 6 blocks of 50 experimental pairs. The order of the pairs within blocks was randomised for each participant, as was the order of the blocks. As a lead in, three filler pairs were presented at the beginning of each block (these were additional to the 150 filler pairs described above), with a 15-second break after each block. The target or filler item would appear first on screen and remain for 200 ms. After a blank screen lasting 50 ms, the probe appeared. The probe remained on screen

[2] A model with a maximal random effects structure converged for all analyses. If a maximal model had not converged, however, the random effects structure would have been reduced in the following manner: (1) the correlations between the random effects were removed, (2) the correlations were reintroduced and the slope that had the smallest variance in the maximal model was removed, (3) the correlations were removed again, (4) the final slope was removed, which left only the random intercepts. This approach differs from *Barr et al.*'s (*2013*) suggestions in that they would recommend to try to remove the random intercepts before removing any random slopes. A model without random intercepts, however, would not take into account the within-participants and within-items nature of this experiment's design.

until the participant responded or until 2,500 ms passed. A warning was presented to the participant that they were responding too slowly if they had not responded 2,000 ms after the probe first appeared. The warning remained on screen for 500 ms, during which time the participant could still respond. The inter-trial interval was 1,000 ms.

## Results

All analyses were carried out in R (version 3.4.3; *R Core Team, 2017*) using the lme4 package (version 1.1-13; *Bates et al., 2015*), following guidelines proposed by *Barr et al. (2013)* for confirmatory hypothesis testing (with some personal amendments[2]) and using Type III Sums of Squares likelihood ratio tests to determine significance of the fixed effects. Reaction times were analysed using the lmer() function with the default optimiser; accuracy data were analysed using the glmer() function with the bobyqa optimiser. Detailed results of all analyses for Experiment 1 can be found in the experimentOverview.xlsx document in our OSF project (http://osf.io/ndb7p/). Finally, the graphs in the figures display the (harmonic) participant means, while the effects reported in the text were derived from the estimates of the fixed effects obtained by calling fixef() on the model.

One item (the cognate "type"–"type") was excluded from the analysis, as its percentage correct for each of the two groups (bilinguals: 58.6%; monolinguals: 50.0%) was more than three standard deviations below its word type mean for both groups ($M_{bilinguals} = 92.6\%$, $SD_{bilinguals} = 8.7\%$; $M_{monolinguals} = 91.7\%$, $SD_{monolinguals} = 8.4\%$). Excluding this item did not affect the matching of the word types. An additional six items (the interlingual homographs "slot"–"slot", "stem"–"stem" and "strand"–"strand" and the English controls "emmer"–"bucket", "lijm"–"glue" and "moeras"–"swamp") were outliers for their word type for either of the two groups, but as per the pre-registration these items were not excluded.

### *Confirmatory analyses*

*Analysis procedure*  The same analysis procedure was employed for the reaction times and accuracy data. Moreover, the same random effects structure converged for both the reaction times analysis and the accuracy analysis. In all cases, positive effects of word type indicate a facilitative effect for the first-named word type over the second (i.e., faster reaction times and higher accuracy), while negative effects indicate inhibitory effects (i.e., slower reaction times and lower accuracy). Positive effects of group indicate an advantage for the bilinguals over the monolinguals, while negative effects indicate an advantage for the monolinguals over the bilinguals.

Two fixed factors were included in the main 3×2 analysis: word type (3 within-participant/ between-items levels: cognate, English control, interlingual homograph) and group (2 between-participant/within-items levels: bilingual, monolingual). The maximal random effects structure included a correlated random intercept and random slope for word type by participants and a correlated random intercept and random slope for group by items.

In addition, a 2×2 analysis similar to the 3×2 analysis was conducted, but which included data for only the cognates and interlingual homographs (for both groups). This analysis allowed us to determine whether the bilinguals and monolinguals treated the cognates

and interlingual homographs differently. The maximal random effects structure for the 2×2 analysis included a correlated random intercept and random slope for word type by participants and a correlated random intercept and random slope for group by items. We also conducted two pairwise comparisons that compared the cognates and interlingual homographs to each other but separately for the bilinguals and the monolinguals. The maximal random effects structure for the pairwise comparisons included a correlated random intercept and random slope for word type by participants and a random intercept by items.

To examine more closely whether specifically there was evidence for a cognate facilitation effect, a 2×2 analysis was conducted which included only the data for the cognates and English controls (for both groups). The maximal random effects structure again included a correlated random intercept and random slope for word type by participants and a correlated random intercept and random slope for group by items. In addition, two pairwise comparisons were conducted, comparing the cognates and English controls to each other but separately for the bilinguals and the monolinguals. The maximal random effects structure for these analyses included a correlated random intercept and random slope for word type by participants and a random intercept by items.

Similarly, to examine more closely whether there was evidence for an interlingual homograph inhibition effect, a 2×2 analysis was conducted which included the data for only the interlingual homographs and English controls (for both groups). The maximal random effects structure again included a correlated random intercept and random slope for word type by participants and a correlated random intercept and random slope for group by items. In addition, two pairwise comparisons were conducted, comparing the interlingual homographs and English controls to each other but separately for the bilinguals and the monolinguals. The maximal random effects structure for these analyses included a correlated random intercept and random slope for word type by participants and a random intercept by items.

For the three 2×2 analyses and the three pairwise comparisons, all $p$-values were compared against a Bonferroni-corrected $\alpha$ of .0167 (i.e., correcting for three comparisons, as each level of analysis was considered a family). As per the pre-registration, the 3×2 analysis and the three 2×2 analyses were each conducted once without age included as a covariate and once with. In most cases, including age as a covariate did not change the significance level of the analysis; in cases when the significance level did change, this will be noted. Finally, age was centred to have a mean of zero prior to including it in the model. Random slopes were not included for age.

*Reaction times* Reaction times are shown in Fig. 1. Trials with reaction times (RTs) less than 300 ms or more than 2,000 ms were discarded (0.5% of the data), as were incorrect trials and trials that participants had not responded to (8.5% of the remaining data). The RTs were inverse-transformed (inverse-transformed RT = 1,000/raw RT), as a histogram of the residuals and a predicted-vs-residuals plot for the main 3×2 analysis showed that the assumptions of normality and homoscedasticity were violated. (The inverse-transform achieved a better distribution of the residuals than a log10-transform.) After transforming
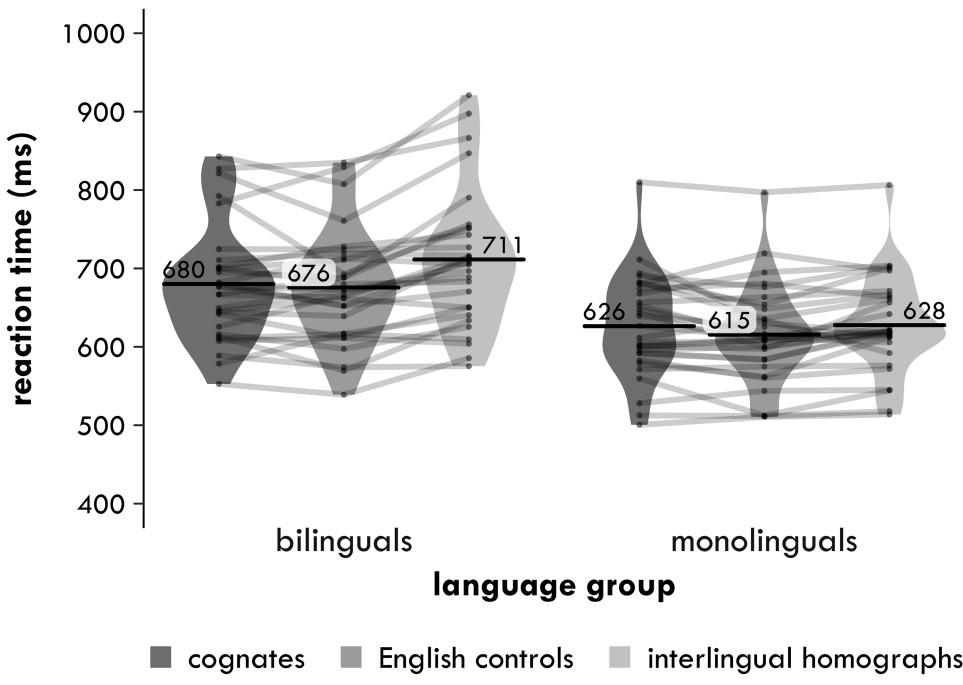

**Figure 1** **Experiment 1. Harmonic participant means of the inverse-transformed semantic relatedness task reaction times (in milliseconds) by group and word type.** Each point represents a condition mean for a participant with lines connecting means from the same participant. Each bar provides the mean across all participants in that condition. The violin is a symmetrical density plot rotated by 90 degrees.

the RTs, any inverse-transformed RTs were removed that were more than three standard deviations above or below a participant's mean inverse-transformed RT (0.2% of the remaining data).

*3 × 2* In the 3×2, the main effect of word type was significant [$\chi^2(2) = 6.068$, $p = .048$], indicating that there was a difference in reaction times between the three word types. The main effect of group was also significant [$\chi^2(1) = 12.74$, $p < .001$], with the bilingual participants responding on average 65 ms more slowly than the monolingual participants. Crucially, the interaction between word type and group was significant [$\chi^2(2) = 7.173$, $p = .028$].

*Cognates versus interlingual homographs* In the 2×2 that included only the cognates and the interlingual homographs, the main effect of word type was not significant [$\chi^2(1) = 2.452$, $p = .117$, $\Delta = 17$ ms]. Crucially, the interaction between word type and group was significant [$\chi^2(1) = 6.072$, $p = .014$]. The main effect of group was also significant [$\chi^2(1) = 12.97$, $p < .001$], with the bilingual participants responding on average 68 ms more slowly than the monolingual participants. The two pairwise comparisons also suggested a difference in how the bilinguals and monolinguals responded to the cognates and the interlingual homographs. The bilinguals responded 35 ms significantly more quickly to the cognates compared to the interlingual homographs [$\chi^2(1) = 6.033$,

$p = .014$], but the 3 ms difference between these word types of for the monolinguals was not significant [$\chi^2(1) = 0.084, p = .772$].

*Cognates versus English controls*  In the 2×2 that included only the cognates and English controls, the main effect of word type was again not significant [$\chi^2(1) = 0.678, p = .411, \Delta = -8$ ms], nor was the interaction between word type and group [$\chi^2(1) = 0.758, p = .384$]. The main effect of group was significant again [$\chi^2(1) = 10.00, p = .002$], with the bilingual participants responding on average 56 ms more slowly than the monolingual participants. The two pairwise comparisons told a similar story. There was no significant advantage for cognates compared to English controls for either the bilinguals [$\chi^2(1) = 0.120, p = .730, \Delta = -4$ ms] or the monolinguals [$\chi^2(1) = 1.334, p = .248, \Delta = -11$ ms].

*Interlingual homographs versus English controls*  In the 2×2 that included only the interlingual homographs and English controls, the main effect of word type was only significant at an uncorrected $\alpha$ of .05 [$\chi^2(1) = 5.711, p = .017, \Delta = -25$ ms]. The interaction between word type and group was not significant [$\chi^2(1) = 3.588, p = .058$], but the main effect of group was again significant [$\chi^2(1) = 14.58, p < .001$], with the bilingual participants responding on average 72 ms more slowly than the monolingual participants. Although the interaction was not significant, the two pairwise comparisons suggested a difference in how the bilinguals and monolinguals responded to the interlingual homographs. The bilinguals responded 39 ms significantly more slowly to the interlingual homographs than the English controls [$\chi^2(1) = 7.915, p = .005$], while for the monolinguals the difference between these two word types was only 14 ms and not significant [$\chi^2(1) = 1.936, p = .164$].

*Accuracy*  Accuracy is shown in Fig. 2. In line with the trimming procedure for the reaction times, any trials with RTs faster than 300 ms or slower than 2,000 ms were removed.

3 × 2  In the 3×2, the main effect of word type was again significant [$\chi^2(2) = 14.00, p = .001$]. The main effect of group was not significant [$\chi^2(1) = 0.029, p = .864, \Delta = 0.2\%$]. Crucially, the interaction between word type and group was again significant [$\chi^2(2) = 6.205, p = .045$]. When age was included as a covariate, however, the interaction became marginally significant [$\chi^2(1) = 5.478, p = .065$]. The effect of age itself was not significant [$\chi^2(1) = 0.491, p = .484$][3].

[3]When age was included, the maximal model would not converge, but a model without correlations between the random effects did.

*Cognates versus interlingual homographs*  In the 2×2 that included the cognates and the interlingual homographs, the main effect of word type was significant [$\chi^2(1) = 10.79, p = .001$], with participants making on average 3.7 percentage points fewer mistakes with the cognates than the interlingual homographs. The interaction between word type and group was also significant [$\chi^2(1) = 5.750, p = .016$], but the main effect of group was not [$\chi^2(1) = 0.108, p = .743, \Delta = -0.4\%$]. Again, the two pairwise comparisons suggest a difference in how the bilinguals and monolinguals processed the cognates and the interlingual homographs. The bilinguals made 5.5 percentage points fewer mistakes

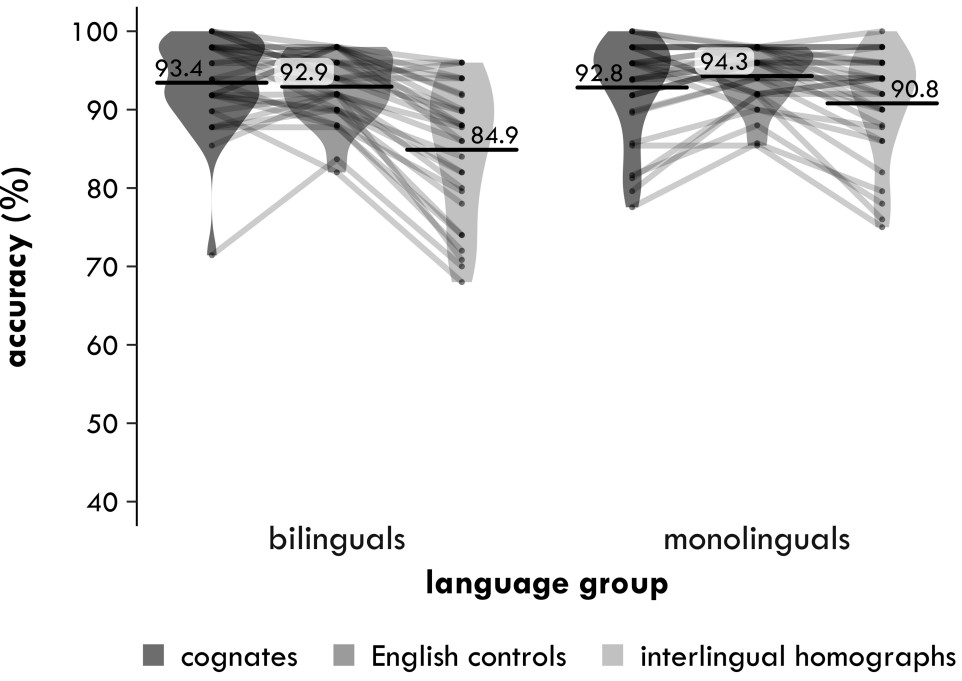

**Figure 2** **Experiment 1. Participant means of semantic relatedness task accuracy (percentages correct) by group and word type.** Each point represents a condition mean for a participant with lines connecting means from the same participant. Each bar provides the mean across all participants in that condition. The violin is a symmetrical density plot rotated by 90 degrees.

with the cognates than the interlingual homographs $[\chi^2(1) = 10.91, p = .001]$, but the difference of 1.8 percentage points for the monolinguals was not significant $[\chi^2(1) = 2.560, p = .110]$.

*Cognates versus English controls* In the 2×2 that included the cognates and English controls, the main effect of word type again was not significant $[\chi^2(1) = 0.110, p = .740, \Delta = 0.2\%]$, nor was the main effect of group $[\chi^2(1) = 0.443, p = .506, \Delta = -0.6\%]$ or the interaction between word type and group $[\chi^2(1) = 1.258, p = .262]$. The two pairwise comparisons told a similar story. There was no significant advantage for cognates compared to English controls for either the bilinguals $[\chi^2(1) = 0.620, p = .431, \Delta = 0.9\%]$ or the monolinguals $[\chi^2(1) = 0.250, p = .617, \Delta = 0.5\%]$.

*Interlingual homographs versus English controls* In the 2×2 that included the interlingual homographs and English controls, in contrast, the main effect of word type was significant $[\chi^2(1) = 9.552, p = .002]$, with participants making on average 3.5 percentage points more mistakes with the interlingual homographs than the English controls. The interaction between word type and group was not significant $[\chi^2(1) = 1.608, p = .205]$, nor was the main effect of group $[\chi^2(1) = 0.294, p = .588, \Delta = 0.7\%]$. As for the reaction times, although the interaction was not significant, the two pairwise comparisons suggest a difference in how the bilinguals and monolinguals processed the interlingual homographs.

The bilinguals made 4.8 percentage points significantly more mistakes with the interlingual homographs than the English controls [$\chi^2(1) = 7.889$, $p = .005$], while the much smaller difference of 2.3 percentage points for the monolinguals was only significant at an uncorrected $\alpha$ of .05 [$\chi^2(1) = 4.085$, $p = .043$].

### Exploratory analyses

An exploratory analysis was conducted to compare the pattern of results observed in this experiment to that found in the lexical decision tasks *Poort & Rodd (2017b)* conducted. As *Poort & Rodd (2017b)* discuss, in lexical decision tasks without an element of response competition (i.e., without non-target language words), cognates tend to elicit facilitation compared to control words, but interlingual homographs are generally not recognised any more slowly (or only a little more slowly) than control words. This is indeed what *Poort & Rodd*'s *(2017b)* found: there was a significant cognate facilitation effect of approximately 31 and 46 ms in the two standard versions of their two experiments, which included only a set of cognates, English controls and non-words. There was only a small, but non-significant interlingual homograph inhibition effect of 8 ms in the +IH version of Experiment 2, which included a set of cognates, English controls, interlingual homographs and non-words. Two 2×2 analyses were conducted to compare these effects to the effects obtained in Experiment 2 of *Poort & Rodd*'s *(2017b)* study, to determine whether the presence and size of the cognate facilitation effect and the interlingual homograph inhibition effect does indeed depend on the type of task (semantic relatedness vs lexical decision).

The first 2×2 analysis focussed on the cognate facilitation effect. This analysis included the semantic relatedness data for the cognates and English controls from the bilingual group of this experiment ($N = 29$) and the lexical decision data from the standard version of *Poort & Rodd*'s *(2017b)* Experiment 2 ($N = 21$). Only the cognates ($n = 49$) and English controls ($n = 49$) that had been included in both experiments were included in this analysis[4]. Two fixed factors were included in this analysis: word type (2 within-participant/between-items levels: cognate, English control) and task (2 between-participant/within-items levels: semantic relatedness, lexical decision). The maximal model converged and included a correlated random intercept and random slope for word type by participants and a correlated random intercept and random slope for task by items. The main effect of task was not significant [$\chi^2(1) = 0.129$, $p = .719$, $\Delta = -9$ ms]. (Positive effects of task indicate faster reaction times in lexical decision than semantic relatedness.) There was a significant main effect of word type [$\chi^2(1) = 9.403$, $p = .002$], with cognates being responded to on average 22 ms more quickly than English controls. Crucially, there was also a significant interaction between word type and task [$\chi^2(1) = 10.93$, $p < .001$], indicating that the cognate facilitation effect was significantly smaller in our semantic relatedness task ($-4$ ms) compared to *Poort & Rodd*'s *(2017b)* lexical decision task (46 ms).

The second 2×2 analysis was conducted in a similar manner as the first but focussed on the interlingual homograph inhibition effect. This analysis included the semantic relatedness data for the interlingual homographs and English controls from the bilingual group of this experiment ($N = 29$) and the lexical decision data for the interlingual homographs and English controls from the +IH version of *Poort & Rodd*'s *(2017b)*

[4]The cognates that were removed were "amber", "ego", "instinct", "jury", "lens", "tennis" and "type". The first six of these items had not been included in Experiment 1 of this paper at all; "type" was included in both experiments but had been excluded from the confirmatory analyses of this experiment. The English controls that were removed were "art", "cruel", "duty", "flu", "poem", "rifle", "sky" and "treaty". The item "sky" had only been included in this experiment, whereas the other items had only been included in *Poort & Rodd*'s *(2017b)* Experiment 2.
[5]The interlingual homographs that were removed were "fee", "hoop", "lever", "mate", "pal" and "toe". These items had all only been included in *Poort & Rodd*'s (*2017b*) Experiment 2. The same English controls were removed for this analysis as for the other analysis.

Experiment 2 ($N = 20$). Again, only the interlingual homographs ($n = 50$) and English controls ($n = 49$) that had been included in both experiments were included in this analysis[5]. As before, two fixed factors were included in this analysis: word type (2 within-participant/between-items levels: interlingual homograph, English control) and task (2 between-participant/within-items levels: semantic relatedness, lexical decision). The maximal model converged again and included a correlated random intercept and random slope for word type by participants and a correlated random intercept and random slope for task by items. The main effect of task was again not significant [$\chi^2(1) = 0.888, p = .346$, $\Delta = 24$ ms]. There was a significant main effect of word type [$\chi^2(1) = 6.967, p = .008$], with interlingual homographs being responded to on average 23 ms more slowly than the English controls. The interaction between word type and task was marginally significant [$\chi^2(1) = 3.303, p = .069$], suggesting that the interlingual homograph inhibition effect was larger in the semantic relatedness task (39 ms) than in *Poort & Rodd*'s (*2017b*) lexical decision task (8 ms).

## Discussion

The aim of Experiment 1 was to examine semantic processing of cognates and interlingual homographs in bilinguals (and monolinguals). In particular, we set out to determine whether the cognate facilitation and interlingual homograph inhibition effects observed in lexical decision tasks are a reflection of how these words are stored in the bilingual mental lexicon or whether these effects could be artefacts of such tasks. In line with our predictions, there was a significant interaction between word type and language group in the main analysis, which indicates that the bilinguals processed the three word types differently than the monolinguals did. Specifically, this interaction between word type and language group was significant in the analysis that included only the cognates and interlingual homographs. The bilingual participants responded 35 ms more slowly and 5.5 percentage points less accurately to the interlingual homographs than to the cognates, while the monolingual participants did not (with effects of 3 ms and 1.8 percentage points).

The other findings were a little less clear-cut, however, which makes it difficult to pinpoint whether this difference between the cognates and interlingual homographs for the bilinguals was due to a facilitation effect for the cognates, an inhibition effect for the interlingual homographs or both. In line with our predictions, the analysis showed that the bilingual participants responded 39 ms significantly more slowly to the interlingual homographs than to the English controls. In contrast, the difference between these two word types for the monolingual participants was only 14 ms and not significant. Despite a difference of 25 ms between these two effects, the interaction between word type and language group was not significant. Similarly, the interaction between word type and language group was not significant in the accuracy analysis, despite the fact that the bilingual participants made 4.8 percentage points more mistakes with the interlingual homographs, while the monolingual participants made only 2.3 percentage points more mistakes.

Regarding the cognates, both groups of participants processed the cognates slightly more slowly than the English controls, in contrast to our predictions. For the monolingual

participants, this difference was 11 ms, while for the bilinguals it was 4 ms. Neither of these effects was significant, however, nor was the interaction between them. The accuracy data told a similar story, although these effects were in the expected direction: the monolingual participants made 0.5 percentage points fewer mistakes with the cognates, while the bilingual participants made 0.9 percentage points fewer mistakes. In short, it seems that the interaction between word type and language group in the main analysis was driven by an interlingual homograph inhibition effect and not a cognate facilitation effect (or both), but the statistics do not quite provide enough evidence to back up this claim.

When comparing the data to the lexical decision data from *Poort & Rodd*'s (*2017b*) Experiment 2, it appeared that the cognate facilitation effect was significantly larger in lexical decision, while the interlingual homograph inhibition effect was marginally significantly bigger in semantic relatedness. This suggests that these two effects are to some extent task dependent. Before discussing the results of Experiment 1 and the comparison with lexical decision in more detail, however, Experiment 2 attempts to replicate these findings in a larger sample, in addition to examining long-term cross-lingual priming of cognates and interlingual homographs.

## EXPERIMENT 2

Experiment 2 was again pre-registered as part of the Center for Open Science's Preregistration Challenge (http://cos.io/prereg/). The stimuli, data and processing and analysis scripts can be found on the Open Science Framework (http://osf.io/2at49/), as well as an Excel document with detailed results from all of the analyses. The preregistration can be retrieved from http://osf.io/y6phs/ (*Poort & Rodd, 2018*). Where applicable, deviations from the pre-registration will be noted.

### Introduction

Experiment 2 had two aims. First, given the surprising lack of a cognate facilitation effect in Experiment 1, to conduct a direct replication of the pattern of results observed in the bilingual group of Experiment 1. And second, given the successful use of a semantic relatedness task in the monolingual domain to replicate effects of long-term priming with homonyms (*Gilbert et al., 2018*), to attempt once again to replicate the cross-lingual long-term priming effect demonstrated by *Poort, Warren & Rodd (2016)*.

As mentioned previously, further evidence in favour of an integrated lexicon comes from studies of cross-lingual long-term priming of cognates and interlingual homographs (e.g., *Bowers, Mimouni & Arguin, 2000*; *Cristoffanini, Kirsner & Milech, 1986*; *Gerard & Scarborough, 1989*; *Poort, Warren & Rodd, 2016*). For example, *Poort, Warren & Rodd (2016)* demonstrated that a single encounter with a cognate or interlingual homograph in Dutch can affect how quickly Dutch–English bilinguals process these words in a subsequent English lexical decision task. They found that if participants had read a Dutch sentence containing a Dutch–English cognate (e.g., "De hond is een gedomesticeerde ondersoort van de **wolf**", "The dog is a domesticated subspecies of the **wolf**"), they responded more quickly to that cognate ("wolf") in a subsequent English lexical decision task than if they had not. In contrast, having encountered an interlingual homograph in a Dutch context

(e.g., "Alleen bijen en wespen hebben een **angel**", "Only bees and wasps have a **sting**") slowed participants down when they later saw that interlingual homograph ("angel") in the English lexical decision task. In other words, experience with a word in one language affected how quickly participants were able to process that word minutes later in another language.

This long-term priming effect was initially demonstrated with ambiguous words in the monolingual domain by *Rodd et al. (2013)* and is thought to be due to a strengthening of the connections between the orthographic representation of a word and its semantic representation at the time of priming. Consequently, when the word is seen again, the meaning that has been encountered most recently is more readily accessible (potentially at the cost of any alternative meanings). Furthermore, based on the assumption that word meanings are represented by patterns of distributed features, *Rodd et al. (2016)* note that it is also possible that priming can lead to changes to the connections between the semantic units of the primed meaning, making it a more stable representation relative to that of the unprimed meaning. The effect of cross-lingual long-term priming observed by *Poort, Warren & Rodd (2016)* offers further evidence in favour of the idea of a shared lexicon for bilinguals. In two follow-up studies, however, *Poort & Rodd (2017a)* did not replicate these findings. Since the (cross-lingual) long-term word-meaning priming effect is largely semantic in nature, this could explain why no effect of priming was found in *Poort & Rodd*'s (*2017a*) experiments: the lexical decision task they used may not have required the participants to access the meanings of the words they saw (although note that *Poort, Warren & Rodd (2016)* also used a lexical decision task).

As mentioned in the Introduction, *Gilbert et al. (2018)* used a semantic relatedness tasks in the monolingual domain to replicate effects of long-term word-meaning priming: if participants had read the sentence "The pig **pen** was muddier than ever" during the first part of their experiment, they responded more quickly to the probe "enclosure" for "pen" than if they had not. Therefore, in addition to replicating the findings from Experiment 1, the aim of Experiment 2 was to investigate, again, whether bilinguals processing cognates and interlingual homographs in their second language are affected by recent experience with these words in their native language. Experiment 2 used the same design again as *Poort, Warren & Rodd (2016)* except instead of using a lexical decision task as the testing task, it used the same semantic relatedness task as Experiment 1. Participants first read Dutch sentences that contained either a cognate (e.g., "wolf"), an interlingual homograph (e.g., "angel") or the Dutch translation of one of the English controls (e.g., "wortel" for "carrot"). For this reason, the English controls will be called translation equivalents from now on. After a brief and non-linguistic filler task that created a delay of approximately 15 min, the participants then completed the semantic relatedness task. Half of the cognates, interlingual homographs and translation equivalents they saw during this final task were items they had come across in a Dutch sentence previously during the experiment.

Our predictions were based on Rodd and colleagues' interpretation of the long-term word-meaning priming effect and *Poort, Warren & Rodd*'s (*2016*) original cross-lingual replication. For the cognates, we predicted that when the bilingual participants encountered such a word in Dutch, this would strengthen the connection between the cognate's shared

form representation and its shared meaning representation. When the participant then saw that same cognate during the English semantic relatedness task, it would be easier to access this stronger meaning representation and to decide it was related to the probe than if the participant had not seen the cognate before. In other words, we predicted that cross-lingual long-term priming would be beneficial for the cognates. For the interlingual homographs, however, the encounter with its Dutch meaning would strengthen the connection between its form representation and its Dutch meaning. This would make it easier during the English semantic relatedness task to access the Dutch meaning again and more difficult to access the English meaning, which would be likely to slow participants down (and make them more prone to errors) when deciding whether that meaning was related to the English probe or not. That is, we predicted that cross-lingual long-term priming would be disruptive for the interlingual homographs. Because the translation equivalents have different written and spoken forms in Dutch and English, encountering the Dutch form of a translation equivalent would be unlikely to interfere with the connection between the English form and the meaning representation of that translation equivalent. As such, we predicted that cross-lingual long-term priming would not affect the translation equivalents. Finally, by analysing the unprimed trials, we could determine whether the pattern of results observed in Experiment 1 replicated (although this time the experiment only included a group of bilingual participants).

## Methods

### Participants

*Power analysis* We conducted a smallest-effect-size-of-interest power analysis for the priming effects in R (version 3.4.3; *R Core Team, 2017*) using the simr package (*Green & MacLeod, 2016*). The simr package calculates power for a particular analysis on a given dataset through Monte Carlo simulations. We used the data from Experiment 1 to run the simulations, post-hoc 'assigning' participants and items to one of the two priming versions as planned for Experiment 2 (see Design and procedure). We then simulated priming effects of approximately 20 ms for the cognates and the interlingual homographs by subtracting or adding a number of milliseconds to the reaction times for the 'primed' items until the difference between the 'primed' and 'unprimed' items was 20 ms. (We did not alter the reaction times for the English controls, which resulted in a 'priming effect' of less than 2 ms.)

A power analysis was conducted only for the analyses that had been specified in the preregistration as being directly related to our hypotheses: the two simple effects analyses of priming for the cognates and the interlingual homographs as well as the two 2×2 analyses that compared the priming effect for the cognates and translation equivalents and for the interlingual homographs and the translation equivalents. Figs. S1 and S2 display the power curves for the simulations of the two simple effects analyses; Figs. S3 and S4 display the power curves for the two 2×2 analyses. Each power curve was based on 50 simulations each for 10 hypothetical sample sizes between 0 and 100. The significance level was set at our planned Bonferroni-corrected level of .0167, as per the preregistration; the reaction times were again inverse-transformed (as the assumptions of normality and homogeneity

were expected to be violated) and a maximal random effects structure was used (on the assumption that this would converge for these analyses). These power curves indicated that at least 80 participants were required to achieve 80% power for the two simple effects analyses and upwards of 100 to achieve 80% power for the two $2 \times 2$ analyses. As it seemed unlikely that we would be able to recruit more than 100 participants, the target was set at 100 participants[6].

*Participant characteristics* The eligibility criteria were the same as for the bilingual participants of Experiment 1. A total of 114 Dutch–English participants was recruited through personal contacts resident in the Netherlands, the University of Ghent SONA system and social media. The participants gave informed consent (by means of ticking a box in the online consent form) and received a gift card worth €10 (roughly £8 at the time) for their participation in the experiment. The UCL Experimental Psychology Ethics Committee provided approval of our study protocol (Project ID: EP/2017/009).

There were again two stages at which participants could be excluded from our analyses. First, while testing was still on-going, participants who scored less than 80% correct on either or both of the semantic relatedness tasks and/or less than 50% on either or both of the two language proficiency measures were excluded and replaced. Participants were also excluded and replaced if their average priming delay (the average delay between the two presentations of each of the primed items) was more than 30 min. Nine participants that met these criteria were excluded and nine new participants tested in their stead. Due to a technical error, the data from four other participants could not be used, so these participants were also replaced. Second, after testing had finished and a total of 101 useable datasets had been gathered, each participant's performance on the targets (cognates, interlingual homographs and translation equivalents) included in the English semantic relatedness task was compared to the grand mean of all participants to determine whether any participants were outliers compared to the rest. All participants had performed within three standard deviations of the mean ($M = 88.1\%$, $SD = 5.6\%$), so none were excluded at this second stage.

The 101 included participants (30 males, 70 females, 1 non-binary; $M_{age} = 24.7$ years, $SD_{age} = 6.6$ years) had started learning English from an average age of 8.1 ($SD = 3.0$ years) and so had an average of 16.6 years of experience with English ($SD = 6.5$ years). The participants rated their proficiency as 9.7 out of 10 in Dutch ($SD = 0.5$) and as 8.7 in English ($SD = 0.9$). A two-sided paired $t$-test showed this difference to be significant [$t(100) = 11.12$, $p < .001$]. These self-ratings were confirmed by their high LexTALE scores in both languages, which a two-sided paired $t$-test showed were also higher in Dutch [Dutch: $M = 88.5\%$, $SD = 6.7\%$; English: $M = 85.8\%$, $SD = 9.3\%$; $t(100) = 2.767$, $p = .007$].

### Materials

The same items were used as for Experiment 1. *Poort & Rodd (2017a)* had written prime sentences in Dutch for these items already (see Table 3 for examples), but minor changes were made to approximately one-fifth of these, for example when an item's probe in

[6] As noted in our pre-registration, we gave ourselves a time frame of 30 days to recruit the required 100 participants, otherwise we would terminate recruitment. Fortunately, that was unnecessary.

**Table 3  Experiment 2. Examples of Dutch prime sentences for a cognate, interlingual homograph and translation equivalent.**

|  | Dutch word form | English word form | prime sentence (Dutch original) | prime sentence (English translation) |
|---|---|---|---|---|
| cognate | wolf | wolf | De hond is een gedomesticeerde ondersoort van de **wolf**. | The dog is a domesticated subspecies of the **wolf**. |
| interlingual homograph | angel | angel | Alleen vrouwelijke bijen en wespen hebben een **angel**. | Only female bees and wasps have a **sting**. |
| translation equivalent | wortel | carrot | Een ezel kun je altijd blij maken met een **wortel**. | You can always make a donkey happy with a **carrot**. |

**Notes.**
The English translations of the sentences were not shown during the experiment.

the English semantic relatedness task was also present in its prime sentence or in any of the other prime sentences. Some sentences were also changed to make sure, as much as possible, that no translations of any of the probes from the English semantic relatedness task occurred in any of the Dutch prime sentences. Furthermore, none of those translations were used as probes in the Dutch semantic relatedness task either.

*Poort & Rodd (2017a)* had also paired each prime sentence with a related or unrelated probe, for use in the Dutch semantic relatedness task (see Design and procedure). These probes were either very strongly related or completely unrelated to the sentence. However, many of the related probes that *Poort & Rodd (2017a)* had originally paired with the prime sentences to use in the Dutch semantic relatedness task were now being used in the English semantic relatedness task. These probes were replaced with new probes. Some prime sentences had to be modified or rewritten entirely if the choice of these new probes was limited. All modified and rewritten sentences were proofread by the same native speaker of Dutch who proofread the sentences for *Poort & Rodd (2017a)*. He also confirmed that the probes were (un)related to the sentences as intended. Independent-samples two-tailed Welch's $t$-tests showed that the differences between the word types in terms of prime sentence length were not significant (all $ps > .6$).

The 15 cognate, 15 interlingual homograph and 15 English control fillers now served a double purpose. First, as in Experiment 1, they were included to ensure that the participants would not assume that any cognate or interlingual homograph always required a "yes"-response. Second, eight of the fillers of each type were primed so that, as primed items that were paired with unrelated probes in the English task, they would also ensure that the participants would not assume that any items they had seen during the Dutch semantic relatedness task would always require a "yes"-response during the English semantic relatedness task.

### Design and procedure

As for Experiment 1, Experiment 2 employed a mixed design. Priming was a within-participants/within-items factor: for each participant, half the targets and eight of the fillers of each word type were primed (i.e., appeared during the priming phase) while half were unprimed (i.e., only occurred in the test phase). Two versions of the experiment were created such that participants saw each experimental item only once but across participants items occurred in both the primed and unprimed conditions. The same fillers were primed

for all participants. Word type was a within-participants/between-items factor: participants saw words from all three word types, but each word of course belonged to only one word type.

The experiment was again created and conducted using the Gorilla Experiment Builder (*Anwyl-Irvine et al., 2019*; http://www.gorilla.sc). The experiment protocol and all tasks are available to preview and copy from Gorilla Open Materials at http://www.gorilla.sc/openmaterials/36777. This time, the participants completed a self-report language background survey in Dutch at the start of the experiment to verify eligibility (see the Participants section for the eligibility criteria). The experiment comprised five separate tasks: (1) the Dutch version of the LexTALE (*Lemhöfer & Broersma, 2012*), (2) the Dutch semantic relatedness task (mean duration in mm:ss: 08:38), (3) the Towers of Hanoi task (with instructions presented in English; maximum duration set to 4 min), (4) the English semantic relatedness task (mean duration: 10:53) and (5) the English version of the LexTALE (*Lemhöfer & Broersma, 2012*). Across participants, English semantic relatedness judgements to primed items were made on average 14 min and 27 s after they were primed in the Dutch semantic relatedness task ($SD = 01:51$; *range* $= 11:37–25:36$). The five tasks were presented separately with no indication that they were linked.

*Dutch semantic relatedness task*  This task served to prime the cognates, interlingual homographs and translation equivalents. To ensure the participants processed the prime sentences, participants were asked to indicate via button presses whether a subsequent probe was semantically related to the preceding sentence. The 50 target sentences for each of the three word types were pseudorandomly divided into two sets of 75, matched for all key variables and prime sentence length, for use in the two versions of the experiment. Including the 24 primed filler items, participants read a total of 101 sentences. Half of the sentences in each version were paired with related probes and half with unrelated probes. A practice block of six sentences was followed by four blocks of 24 or 25 sentences (mixed targets and fillers). The order of the items within a block was randomised for each participant, as was the order of the blocks. Of all the items of each word type that were assigned to a block (6 or 7 items), 1 (or 2) item(s) was/were assigned to each of the six blocks of the English semantic relatedness task. This was to ensure that the (variation in) duration of the delay would be similar for each of the three word types. A 15-second break was enforced after each block and three fillers (additional to the 24 cognate, interlingual homograph and translation equivalent fillers) were presented at the start of each block. Participants were allowed to read the sentences at their own pace, with a minimum presentation time of 1,000 ms and a maximum of 4,000 ms. Participants pressed the spacebar after they had read the sentence, after which the probe appeared on the screen. Each probe remained on the screen until the participant responded or until 2,000 ms passed. The inter-trial interval was 500 ms.

*Towers of Hanoi task*  This task served to introduce a delay between priming and testing, while minimising exposure to additional linguistic material. The Towers of Hanoi is a puzzle in which disks of progressively smaller sizes must be moved from one peg to

another. There are two simple rules: (1) only one disk may be moved at a time and (2) a larger disk may not be placed on top of a smaller disk. The goal is to move the disks from the starting peg to the finish peg in as few moves as possible. Participants were given 4 min to complete as many puzzles as they could, starting with a puzzle with three disks and three pegs. Each subsequent puzzle had the same number of pegs but one disk more than the previous puzzle. The instructions were presented in English to minimise any general language switching cost on the English semantic relatedness task.

*English semantic relatedness task* The English semantic relatedness task was identical to that used for Experiment 1, except that participants were given only 2,000 ms to respond, to reduce the overall duration of this task and of the whole experiment. A warning was presented to the participant that they were responding too slowly if they had not responded 1,500 ms after the probe first appeared. The warning remained on screen for 500 ms, during which time the participant could still respond.

## Results

All analyses were again carried out in R (version 3.4.4; *R Core Team, 2018*) using the lme4 package (version 1.1–17; *Bates et al., 2015*), following *Barr et al.*'s (*2013*) guidelines (with the same personal amendments as in Experiment 1) and using Type III Sums of Squares likelihood ratio tests to determine significance. Reaction times were analysed using the lmer() function with the default optimiser; accuracy data were analysed using the glmer() function with the bobyqa optimiser. Detailed results of all analyses for Experiment 2 can be found in the experimentOverview.xlsx document in our OSF project (http://www.osf.io/2at49/). Again, the graphs in the figures display the (harmonic) participant means, while the effects (and marginal means, *MM*) reported in the text were derived from the estimates of the fixed effects provided by the model summary. Finally, this section reports the results of the planned confirmatory analyses. No exploratory analyses are reported.

Four items (the cognate "nest"–"nest", the interlingual homograph "slot"–"slot" and the translation equivalents "emmer"–"bucket" and "moeras"–"swamp") were excluded from the analyses, as the percentages correct on the English semantic relatedness task for those items (61.4%, 23.8%, 35.6% and 55.4%, respectively) were more than three standard deviations below the items' word type mean (for the cognates: $M = 92.1\%$, $SD = 7.8\%$; for the interlingual homographs: $M = 81.3\%$, $SD = 17.9\%$; for the translation equivalents: $M = 90.8\%$, $SD = 11.6\%$). Excluding these items did not affect the matching of the word types.

### Towers of Hanoi task

Every participant completed at least one puzzle, confirming task engagement. On average, participants completed 2.3 puzzles (*mode = 2, range = 1–3*).

### Dutch semantic relatedness task

High accuracy ($M = 93.6\%$, $SD = 3.9\%$, *range = 80.8%–99.0%*) confirmed participants had processed the sentence meanings. To determine whether any of the observed effects of

priming in the English semantic relatedness task could have been due to differences between the word types or priming versions at the time of priming, a 3×2 analysis was conducted on the accuracy data with the fixed factors word type (3 within-participants/between-items levels: cognate, interlingual homograph, translation equivalent) and priming version (2 between-participants/within-items levels: version 1, version 2). The maximal model converged for this analysis and included a random intercept by participants and items as well as a by-participants random slope for word type and a by-items random slope for version. This analysis revealed that the main effect of word type was not significant [$\chi^2(2) = 1.226$, $p = .542$; $MM_{\text{cognates}} = 97.6\%$, $MM_{\text{interlingual homographs}} = 98.2\%$, $MM_{\text{translation equivalents}} = 97.6\%$] but the main effect of version was [$\chi^2(1) = 5.824$, $p = .016$], with participants in version 1 ($MM = 98.4\%$) outperforming those in version 2 ($MM = 96.8\%$) by 1.6 percentage points on average. Note that this difference between the two versions could not have affected any of our key findings, which concern the differences between priming for the different word types. The interaction was also not significant [$\chi^2(1) = 2.999$, $p = .223$].

As per our pre-registration, three pairwise comparisons were also conducted, to directly compare the cognates, interlingual homographs and translation equivalents to each other. The maximal model also converged for these three analyses and included a random intercept by participants and items and a by-participants random slope for word type. The $p$-values for the pairwise comparisons were compared against a Bonferroni-corrected $\alpha$ of .0167. Again, these analyses revealed no significant differences between the word types [cognates ($MM = 97.5\%$) vs translation equivalents ($MM = 97.9\%$): $\chi^2(1) = 0.240$, $p = .625$; interlingual homographs ($MM = 98.1\%$) vs translation equivalents ($MM = 97.5\%$): $\chi^2(1) = 0.704$, $p = .402$; cognates ($MM = 97.1\%$) vs interlingual homographs ($MM = 98.2\%$): $\chi^2(1) = 1.854$, $p = .173$].

### English semantic relatedness task

*Analysis procedure* The same analysis procedure was employed for the reaction times and accuracy data. In all cases, positive effects of priming indicate a facilitative effect of priming (i.e., faster reaction times and higher accuracy for primed items), while negative effects indicate a disruptive effect of priming (i.e., slower reaction times and lower accuracy for primed items). Positive (negative) effects of word type indicate an advantage (disadvantage) for the first-named word type over the second-named word type.

Two fixed factors were included in the main 2×3 analysis: priming (2 within-participant/within-items levels: unprimed, primed) and word type (3 within-participant/between-items levels: cognate, interlingual homograph, translation equivalent). The maximal random effects structure of this model included a correlated random intercept and random slope for word type, priming and their interaction by participants and a correlated random intercept and random slope for priming by items. This maximal model converged for the accuracy analysis but not the reaction time analysis, for which a model without correlations between the random effects converged.

In addition, three 2×2 analyses were conducted comparing the effect of priming for the cognates and interlingual homographs, the cognates and translation equivalents

and the interlingual homographs and translation equivalents. The maximal random effects structure for these models also included a correlated random intercept and random slope for word type, priming and their interaction by participants and a correlated random intercept and random slope for priming by items. Again, the maximal model only converged for the accuracy analysis; a model without correlations between the random effects converged for the reaction times analysis. The $p$-values for these three analyses were compared against a Bonferroni-corrected $\alpha$ of .0167.

To examine the effect of priming for each of the three word types separately, three simple effects analyses were conducted. The maximal model converged for both the reaction times and accuracy analysis and included a correlated random intercept and random slope for priming by both participants and items. The $p$-values for these three analyses were compared against a Bonferroni-corrected $\alpha$ of .0167.

Finally, three pairwise comparisons were conducted on the unprimed data only, comparing the cognates, interlingual homographs and translation equivalents to each other as in Experiment 1. Only the unprimed trials were analysed as priming was expected to essentially increase both the cognate facilitation effect and the interlingual homograph inhibition effect[7], so this would not be a fair test of replication. The maximal model converged for these three models for both the reaction times and accuracy analysis and included a correlated random intercept and random slope for word type by participants and a random intercept by items. The $p$-values for these three analyses were compared against a Bonferroni-corrected $\alpha$ of .0167.

[7]Because primed cognates were predicted to be responded to more quickly than unprimed cognates and primed interlingual homographs more slowly than unprimed interlingual homographs, while priming was predicted to be ineffective for the translation equivalents.

*Reaction times* Reaction times are shown in Fig. 3. As in Experiment 1, reaction times (RTs) faster than 300 ms or slower than 1,500 ms were discarded (0.9% of the data), as were RTs for incorrect trials and trials that participants had not responded to (10.4% of the remaining data). The RTs were again inverse-transformed (inverse-transformed RT = 1,000/raw RT) as a histogram of the residuals and a predicted-vs-residuals plot for the main 2×3 analysis showed that the assumptions of normality and homoscedasticity were violated. After inverse-transforming the RTs, any inverse-transformed RTs were removed that were more than three standard deviations above or below a participant's mean inverse-transformed RT (0.1% of the remaining data).

_2 × 3_  In the 2×3 analysis, the main effect of priming was not significant $[\chi^2(1) = 0.115, p = .734, \Delta = -1 \text{ ms}]$. The main effect of word type was significant, however $[\chi^2(2) = 24.66, p < .001]$, indicating that there was a difference in reaction times between the three word types. Critically, the interaction between word type and priming was also significant $[\chi^2(2) = 6.703, p = .035]$.

_2 × 2s_  In the 2×2 analysis that included only the cognates and interlingual homographs, the main effect of priming was not significant either $[\chi^2(1) = 0.480, p = .488, \Delta = -2 \text{ ms}]$. The main effect of word type was significant again $[\chi^2(1) = 19.37, p < .001]$, with participants responding on average 49 ms more quickly to the cognates than the interlingual homographs. Crucially, the interaction between word type and priming was significant $[\chi^2(1) = 6.412, p = .011]$. In the 2×2 analysis that included only the

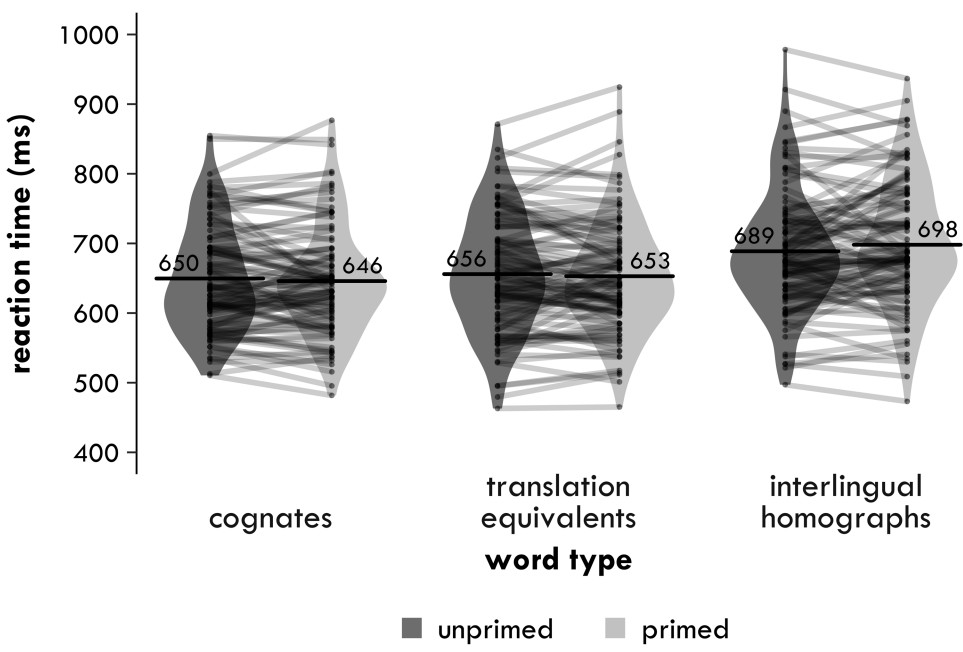

**Figure 3** **Experiment 2. Harmonic participant means of the inverse-transformed English semantic relatedness task reaction times (in milliseconds) by word type and priming status.** Each point represents a condition mean for a participant with lines connecting unprimed and primed means from the same participant. Each bar provides the mean across all participants in that condition. The violin is a symmetrical density plot rotated by 90 degrees.

cognates and translation equivalents, the main effect of priming was again not significant $[\chi^2(1) = 1.725, p = .189, \Delta = 3$ ms], nor was the main effect of word type $[\chi^2(1) = 0.280, p = .597, \Delta = 5$ ms] or the interaction between word type and priming $[\chi^2(1) = 0.278, p = .605]$. Finally, in the 2×2 analysis that included only the interlingual homographs and translation equivalents, the main effect of priming was also not significant $[\chi^2(1) = 1.707, p = .191, \Delta = -4$ ms] nor was the interaction between word type and priming $[\chi^2(1) = 3.234, p = .072]$. The main effect of word type was significant $[\chi^2(1) = 17.26, p < .001]$, with participants responding on average 44 ms more slowly to the interlingual homographs than the translation equivalents.

*Simple effects* In line with these findings, none of the simple effects of priming were significant at the Bonferroni-corrected $\alpha$ of .0167, though the effect of priming was significant at an uncorrected $\alpha$ of .05 for the interlingual homographs [for the cognates: $\chi^2(1) = 1.739, p = .187, \Delta = 5$ ms; for the interlingual homographs: $\chi^2(1) = 4.334, p = .037, \Delta = -10$ ms; for the translation equivalents: $\chi^2(1) = 0.308, p = .579, \Delta = 2$ ms].

*Pairwise comparisons* The pairwise comparisons on the unprimed trials revealed a significant disadvantage of 37 ms for the interlingual homographs compared to the English controls $[\chi^2(1) = 11.72, p = .001]$, but no significant advantage for the cognates
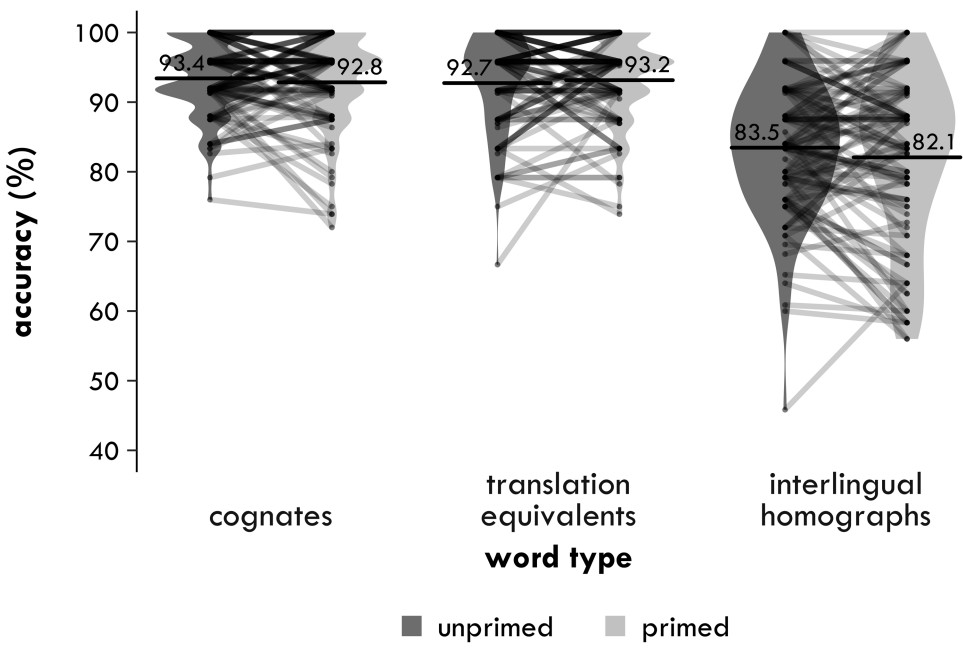

**Figure 4** **Experiment 2. Participant means of English semantic relatedness task accuracy (percentages correct) by word type and priming status.** Each point represents a condition mean for a participant with lines connecting unprimed and primed means from the same participant. Each bar provides the mean across all participants in that condition. The violin is a symmetrical density plot rotated by 90 degrees.

$[\chi^2(1) = 0.149, p = .699, \Delta = 4 \text{ ms}]$. There was also a significant difference between the cognates and interlingual homographs of 41 ms $[\chi^2(1) = 13.34, p < .001]$. These results are fully consistent with the bilingual data from Experiment 1.

*Accuracy* Accuracy is shown in Fig. 4. In line with the trimming procedure for the reaction times, any trials with RTs faster than 300 ms or slower than 1,500 ms were removed.

*2 × 3* In the 2×3 analysis, the main effect of priming was not significant $[\chi^2(1) = 0.194, p = .660, \Delta = 0.2\%]$, but the main effect of word type was $[\chi^2(2) = 20.99, p < .001]$. The interaction between word type and priming was also not significant $[\chi^2(2) = 0.254, p = .881]$.

*2 × 2s* In the 2×2 analysis that included only the cognates and interlingual homographs, the main effect of priming was not significant $[\chi^2(1) = 0.028, p = .868, \Delta = 0.1\%]$, nor was the interaction between word type and priming $[\chi^2(1) = 0.131, p = .717]$. The main effect of word type was significant $[\chi^2(1) = 13.45, p < .001]$, with participants making on average 6.2 percentage points fewer mistakes with the cognates than the interlingual homographs. In the 2×2 analysis that included only the cognates and translation equivalents, the main effect of priming was again not significant $[\chi^2(1) = 0.340, p = .560, \Delta = 0.3\%]$, nor was the main effect of word type $[\chi^2(1) = 0.0002, p = .990, \Delta = -0.01\%]$, or the interaction between word type and priming $[\chi^2(1) = 0.005, p = .946]$. Finally, in the 2×2 analysis that

included only the interlingual homographs and translation equivalents, the main effect of priming was also not significant [$\chi^2(1) = 0.062$, $p = .803$, $\Delta = 0.2\%$], nor was the interaction between word type and priming [$\chi^2(1) = 0.088$, $p = .767$]. The main effect of word type was significant [$\chi^2(1) = 13.97$, $p < .001$], with participants making on average 6.2 percentage points more mistakes with the interlingual homographs than the English controls.

*Simple effects*  In line with these findings, none of the simple effects of priming were significant [for the cognates: $\chi^2(1) = 0.098$, $p = .755$, $\Delta = 0.2\%$; for the interlingual homographs: $\chi^2(1) = 0.022$, $p = .883$, $\Delta = -0.2\%$; for the translation equivalents: $\chi^2(1) = 0.121$, $p = .728$, $\Delta = 0.3\%$].

*Pairwise comparisons*  Again, in line with the results from Experiment 1, the pairwise comparisons on the unprimed trials revealed a significant disadvantage of 6.5 percentage points for the interlingual homographs compared to the English controls [$\chi^2(1) = 13.80$, $p < .001$], but no significant advantage for the cognates [$\chi^2(1) = 0.059$, $p = .808$, $\Delta = -0.3\%$]. There was also a significant difference between the cognates and interlingual homographs of 6.3 percentage points [$\chi^2(1) = 12.76$, $p < .001$].

## Discussion

The aim of Experiment 2 was two-fold: (1) to attempt to replicate the pattern of results in the bilingual group from Experiment 1, who showed an effect of interlingual homograph inhibition but no effect of cognate facilitation and (2) to determine whether cross-lingual long-term priming affects bilingual language processing. In terms of the first aim, the analysis of the unprimed trials of Experiment 2 revealed the same pattern of results as Experiment 1: significant disadvantages of 37 ms and 6.5 percentage points for the interlingual homographs compared to the English controls (i.e., the unprimed translation equivalents) but non-significant differences of only 4 ms and 0.3 percentage points between the cognates and the English controls. (Note that, although the reaction time effect for the cognates was still not significant, it was now in the expected direction. The small effect in the accuracy data was in the opposite direction to what was predicted and found in Experiment 1). Also consistent with the results from Experiment 1, the interlingual homographs were again processed more slowly and less accurately than the cognates, by 41 ms and 6.3 percentage points. In short, these results indicate that the findings from Experiment 1 are reliable.

In terms of the second aim, the evidence for an effect of cross-lingual long-term priming was weak. In the reaction time data, there was a significant interaction between word type and priming in the main analysis, as predicted. Crucially, the effect of priming was significantly different for the cognates compared to the interlingual homographs. As in Experiment 1, however, the other findings are less straightforward, making it difficult to determine whether this difference was due to a facilitatory effect of priming for the cognates or a disruptive effect for the interlingual homographs. Although the participants responded 5 ms more quickly to primed cognates than unprimed cognates, this effect was not significant and much smaller than the *a priori* effect size of interest (20 ms). Neither was

the 10 ms disruptive effect of priming significant for the interlingual homographs (though note that it was significant at an uncorrected $\alpha$ of .05). In line with our predictions, the 2 ms 'facilitative' effect for the translation equivalents was also not significant. Unsurprisingly given the sizes of the effects, but again in contrast to the predictions, the priming effects for the cognates and for the interlingual homographs were not significantly different from the effect for the translation equivalents. Finally, there were no effects of priming in the accuracy data at all. In sum, despite using a task that required the participants to access the meanings of the words they saw and so was predicted to show a priming effect, a strong effect of priming was not observed. This is striking, as research in the monolingual domain has shown an effect of long-term priming when using a similar task (*Betts, 2018*; *Gilbert et al., 2018*). The findings of Experiment 2 will be discussed in more detail along with the findings of Experiment 1 in the General discussion.

# GENERAL DISCUSSION

## Processing cognates and interlingual homographs: task effects

One of the aims of the experiments presented here was to examine how bilinguals process cognates and interlingual homographs in a task that requires semantic access. In line with previous research using mainly lexical decision tasks, both Experiment 1 and 2 showed that bilinguals processed cognates differently than interlingual homographs. Furthermore, this was not the case for the monolinguals in Experiment 1, which indicates that the bilinguals processed these words differently because of their knowledge of them in Dutch and not because of pre-existing differences between the cognates and interlingual homographs in the relatedness of their probes. Specifically, when bilingual participants were asked to indicate whether a pair of words was semantically related or not, pairs that included an interlingual homograph were responded to 30–40 ms more slowly and elicited 5–6 percentage points more errors than pairs that included a cognate.

This interaction appeared to be due only to an interlingual homograph inhibition effect and not a cognate facilitation effect. The bilingual participants were 37–39 ms slower on trials that included an interlingual homograph than on trials that included an English control and made approximately 5–6 percentage points more mistakes on these trials. While the exploratory analysis of Experiment 1 hinted that this effect was greater than the effect we observed in a previous lexical decision task that used virtually the same set of items (*Poort & Rodd, 2017b*), the interaction with the monolingual participants did not reach significance, so some caution is advised when interpreting this finding. Unexpectedly, the experiments did not indicate that the bilinguals found the cognates easier to process than the English controls, which may call into question how researchers traditionally view these words. Indeed, the exploratory analysis of Experiment 1 demonstrated that the cognate facilitation effect we had previously found in a lexical decision task (*Poort & Rodd, 2017b*) was significantly larger than the small disadvantage for the cognates that we observed in this semantic relatedness task.

The presence of an interlingual homograph inhibition effect fits in with previous research showing such an effect for interlingual homographs in lexical decision tasks (*Dijkstra,*

*Van Jaarsveld & Ten Brinke, 1998*; *Dijkstra, Grainger & Van Heuven, 1999*; *Lemhöfer & Dijkstra, 2004*; *Van Heuven et al., 2008*), sentence processing (*Libben & Titone, 2009*; *Titone et al., 2011*), auditory word recognition (*Schulpen et al., 2003*; *Lagrou, Hartsuiker & Duyck, 2011*) and word production (*Jared & Szucs, 2002*; *Smits et al., 2006*). Furthermore, this finding is in line with research conducted by *Degani, Prior & Hajajra (2018)*, *Prior et al. (2017)* and *Macizo, Bajo & Cruz Martín (2010)* who also used a semantic relatedness task to investigate the processing of interlingual homographs. As mentioned in the Introduction, in their experiments, bilingual participants were slower to respond to interlingual homophones/-graphs that were paired with probes that were related only to the non-target language meanings of the interlingual homophones/-graphs than to control words paired with the same unrelated probe.

The fact that an interlingual homograph inhibition effect was observed in a semantic relatedness task also appears to fit in the framework of the BIA+ model of the bilingual mental lexicon (*Dijkstra & Van Heuven, 2002*). (Note that the Multilink model (*Dijkstra et al., 2018*) does not currently offer an account of interlingual homographs, as discussed in the Introduction.) This model claims that interlingual homographs consist of two identical orthographic nodes that are each connected to their own semantic node (*Kerkhofs et al., 2006*). The lateral inhibition between these two orthographic nodes could have been the cause of the disadvantage for interlingual homographs observed in the semantic relatedness task used in this experiment, on the assumption that this would last long enough to survive the processing of the probe, which was presented after the interlingual homograph. It is less clear whether the BIA+ model would also be able to account for the finding that the interlingual homograph inhibition effect was somewhat larger in our semantic relatedness task than in *Poort & Rodd*'s (*2017b*) lexical decision task.

The lack of a cognate benefit in this task seems inconsistent with the large body of research that suggests that the cognate facilitation effect is very robust and transfers across tasks. As discussed, the cognate facilitation effect has been observed mainly in visual lexical decision tasks (*Cristoffanini, Kirsner & Milech, 1986*; *De Groot & Nas, 1991*; *Dijkstra, Grainger & Van Heuven, 1999*; *Dijkstra et al., 2010*; *Dijkstra, Van Jaarsveld & Ten Brinke, 1998*; *Font, 2001*; *Lemhöfer & Dijkstra, 2004*; *Lemhöfer et al., 2008*; *Peeters, Dijkstra & Grainger, 2013*; *Sánchez-Casas, García-Albea & Davis, 1992*; *Van Hell & Dijkstra, 2002*), but it has also been found in word production tasks like picture naming (e.g., *Costa, Caramazza & Sebastián-Gallés, 2000*) and single-word reading out loud (e.g., *Schwartz, Kroll & Diaz, 2007*). It has even been found when cognates were embedded in sentences during single sentence reading (*Schwartz & Kroll, 2006*; *Duyck et al., 2007*; *Van Hell & De Groot, 1998*; *Libben & Titone, 2009*; *Van Assche et al., 2009*) and during reading of an entire novel (*Cop et al., 2016*), although the facilitation effect is (much) smaller in these studies.

With respect to the semantic relatedness literature in the bilingual domain, the findings regarding cognates are mixed. Our results are in line with the study conducted by *Yudes, Macizo & Bajo (2010)*, who found that Spanish–English bilingual participants completing a semantic relatedness task in Spanish did not respond more quickly or slowly to word pairs that included a cognate than to word pairs that did not. Our findings are in contrast,

however, to those of *Degani, Prior & Hajajra (2018)*, whose Arabic–Hebrew participants did respond more accurately and marginally more quickly to cognates than to control words in a Hebrew semantic relatedness task. It is unclear where these differences in results come from, as the three studies varied greatly. For example, *Yudes, Macizo & Bajo (2010)* used almost exclusively non-identical cognates, while we used exclusively identical cognates and *Degani, Prior & Hajajra (2018)* examined different-script bilinguals. Further research would be necessary to shed light on this issue.

Perhaps more problematically, however, our findings also do not seem to fit in the framework of the BIA+ model (*Dijkstra & Van Heuven, 2002*) or the newer Multilink model (*Dijkstra et al., 2018*). In the BIA+ model, cognates, contrary to interlingual homographs, consist of a single orthographic node that is connected to a single semantic node (*Peeters, Dijkstra & Grainger, 2013*); in the Multilink model (*Dijkstra et al., 2018*) they consist of two orthographic nodes connected to a shared semantic node. Feedback looping back and forth from the semantic node to the orthographic node(s) (and extra activation from the second orthographic node in the Multilink model) is thought to be the cause of the cognate facilitation effect, as this feedback loop increases the activation level of cognates more quickly than for any other word. It is unclear exactly how the BIA+ and Multilink models would incorporate the lack of a cognate facilitation effect in these experiments. One way would be to allow for separate (but identical) semantic representations for a cognate, one for each language, but then the models would need to provide an alternative account for the facilitation effect observed in the tasks mentioned above.

To sum up, so far, Experiment 1 and 2 both suggest a disadvantage for the interlingual homographs compared to the English control words in a semantic relatedness task, but no difference in the processing of the cognates and English controls. Based on the assumption that the cognate facilitation and interlingual homograph inhibition effect often seen in lexical decision reflect lexical processing and are not merely task artefacts, we predicted both an interlingual homograph inhibition effect and a cognate facilitation effect. Instead, as the exploratory analysis of Experiment 1 showed, the cognate facilitation effect we found here was significantly smaller than the effect we found previously in a lexical decision task (*Poort & Rodd, 2017b*) and the interlingual homograph inhibition effect that we have demonstrated now was bigger than the effect *Poort & Rodd (2017b)* found previously, although this interaction was only marginally significant.

Before we discuss our preferred interpretation of these findings, it should be noted that the semantic relatedness task, like any task, involves decision processes. Consequently, the effects (or lack thereof) that we observed could have been due to response competition, especially for the interlingual homographs. As discussed in the Introduction, whether this response competition would have been caused by language membership information or by information about the degree of relatedness between the target and probe is unclear. Specifically, when responding to pairs that included an interlingual homograph (e.g., "angel"–"heaven"), the fact that the unrelated meaning of the interlingual homograph (i.e., "insect's string") was the *Dutch* meaning of the interlingual homograph may have tempted the participants to respond "no". However, they may also have been tempted to respond "no" in this case solely because "insect's sting" is not semantically related to "heaven". The

latter seems more likely, as Experiment 1 was advertised and conducted entirely in English so that the participants would not assume that their knowledge of the stimuli in Dutch would be relevant to the task. Indeed, only four of the 29 bilingual participants indicated that they had assumed that their knowledge of Dutch would be relevant (and another five said they had suspected so). Furthermore, unlike in the experiment conducted by *Degani, Prior & Hajajra (2018)*, *Prior et al. (2017)* and *Macizo, Bajo & Cruz Martín (2010)*, none of the probes in this experiment were exclusively related to the Dutch meanings of the interlingual homographs.

In other words, it is possible that the interlingual homograph inhibition effect observed here was (partially) the result of response competition. In that case, however, it seems more likely that this response competition was lexico-semantic in nature and arose because the "insect's sting" meaning was unrelated to the concept of "heaven" and competed with the related "spiritual being" meaning. It seems less likely that response competition emerged at the decision stage because the "insect's sting" meaning was known to be the Dutch meaning and participants had strategically linked the Dutch meanings of the interlingual homographs to the "no"-response. Further evidence for this comes from the fact that the interlingual homograph inhibition effect was not bigger or smaller in Experiment 2 than in Experiment 1, despite the fact that participants in Experiment 2 were aware that their knowledge of Dutch was relevant to the experiment. This indicates that participants did not rely on language membership information during the semantic relatedness task and that any response competition that may have arisen was really due to competition between the two meanings of the interlingual homographs.

Furthermore, although response competition may have played a role in the interlingual homograph inhibition effect, it seems unlikely that it could account for the absence of a cognate facilitation effect. For the cognates, which share their meaning across languages, the correct response would always have been "yes", as the probe was related to the cognate's meaning in both Dutch and English. Although there may be variation in how the cognates in this experiment are used in Dutch and English in daily life, the probes were not chosen to be related only to a specific sense of the cognates in English that they did not share with Dutch. It is, therefore, implausible that response competition (partially) cancelled out a facilitation effect as in the lexical decision tasks conducted by *Poort & Rodd (2017b)*.

Instead, research from the monolingual literature may be able to explain these contradicting patterns of facilitation and inhibition for cognates and interlingual homographs in lexical decision tasks and semantic relatedness tasks. Using lexical decision tasks, many studies in the monolingual domain (e.g., *Beretta, Fiorentino & Poeppel, 2005*; *Klepousniotou & Baum, 2007*; *Rodd, Gaskell & Marslen-Wilson, 2002*) have shown that polysemous words—words with many related senses like "twist", which means "to make into a coil", "to operate by turning", "to misconstrue the meaning of", etc.—are often processed more quickly than unambiguous words like "dance". In contrast, in such a task there is often no difference in processing times or only a small disadvantage for homonyms—words with multiple unrelated meanings like "bark", which means either "the cover of a tree" or "the sound a dog makes"—compared to unambiguous words. In tasks that involve more semantic processing, however, homonyms are often associated

with a processing disadvantage (*Hino, Pexman & Lupker, 2006*), while polysemous words are not processed any more quickly or slowly than unambiguous control words.

There is a remarkable similarity between the pattern of results described for homonyms and polysemes in these monolingual studies and the contradiction between the pattern of results observed for the cognates and interlingual homographs in this experiment and the pattern of results found in the lexical decision tasks in Experiment 2. In the monolingual domain, the disadvantage for homonyms is often only small in lexical decision tasks, as it is for interlingual homographs in the bilingual domain when those tasks do not include non-target language words. In tasks that involve more semantic processing, homonyms often show a strong disadvantage, as the interlingual homographs did in this experiment. Indeed, as discussed, the exploratory analysis for Experiment 1 revealed that the interlingual homograph inhibition effect was larger in the semantic relatedness task used in Experiment 1 than in the lexical decision task used in the +IH version of Experiment 2, although this interaction was only marginally significant. This striking similarity between the monolingual and bilingual domain suggests that interlingual homographs are not just etymologically the bilingual equivalent of homonyms, but also behaviourally.

Similarly, there is a compelling resemblance between the processing of polysemous words by monolinguals and the processing of cognates by bilinguals. In lexical decision tasks, monolinguals often process polysemes more quickly than unambiguous words, as bilinguals process cognates more quickly than control words (at least when the task does not include non-target language words). In tasks that require access to a specific meaning or sense of a word, polysemes are not processed differently than unambiguous words in the monolingual domain, the same way cognates were not processed differently than the English controls in this bilingual experiment. As previously mentioned, the exploratory analysis for Experiment 1 revealed that the cognate facilitation effect was larger in the lexical decision task used in the standard version of Experiment 2 than in the semantic relatedness task used Experiment 1. This suggests that cognates may be processed by bilinguals and represented in the bilingual mental lexicon in a similar manner as polysemous words are processed by monolinguals and represented in the monolingual lexicon.

In the monolingual domain, *Rodd, Gaskell & Marslen-Wilson (2004)* developed a distributed connectionist model to account for the effects of different types of ambiguity (homonymy and polysemy) on processing times in lexical decision tasks. In this distributed connectionist model, homonyms consist of a single orthographic pattern that is connected to two separate semantic patterns. These semantic patterns form deep and narrow attractor basins in different regions of the semantic space. Settling into one of these attractor basins is difficult for the model and this is thought to cause the disadvantage for homonyms. Polysemous words, in contrast, consist of a single orthographic pattern that is connected to multiple overlapping semantic patterns. These patterns are close together in semantic space and, because of this, they form a broad and shallow attractor basin with multiple stable states. It is relatively easy for the network to settle into such an attractor basin and this results in a processing advantage for polysemous words compared to unambiguous words.

*Armstrong & Plaut (2008)* note that the settling dynamics of such a distributed connectionist model can also account for the different patterns of facilitation and inhibition observed for homonyms and polysemes in different kinds of tasks (but see *Hino, Pexman & Lupker, 2006*, for an account of these findings in terms of decision-making processes). Indeed, *Rodd, Gaskell & Marslen-Wilson*'s (*2004*) simulations already revealed that in the later stages of settling, there was an advantage for the unambiguous words over the polysemous words. This suggests that in tasks in which access to a specific sense of the word is required, such as semantic categorisation or a semantic relatedness task, there may not be an advantage for polysemous words. *Armstrong & Plaut*'s (*2008*) model further showed that tasks that require little semantic activation would show a polysemy advantage but no homonymy disadvantage, while tasks that require high levels of semantic activation and specifically the retrieval of a precise semantic pattern would show no effect of polysemy but only a homonymy disadvantage. Using a lexical decision task with different levels of non-word difficulty, which determines the level of semantic specificity that is required, they supported the predictions from their computational model: when the non-words were easy and participants did not need to access a specific semantic representation of the words, a polysemy advantage was found; when the non-words were difficult and participants did have to retrieve a specific semantic representation, a homonymy disadvantage was observed.

In sum, in the monolingual domain, the differences in how participants process homonyms and polysemes in different tasks appear best accounted for by the settling dynamics of a distributed connectionist network (*Armstrong & Plaut, 2008*; *Armstrong & Plaut, 2016*). It seems, then, that such a model may also be more appropriate to explain the findings from Experiment 1 than a localist connectionist model like the BIA+ (*Dijkstra & Van Heuven, 2002*) or the Multilink model (*Dijkstra et al., 2018*). We propose that in such a bilingual distributed connectionist model, interlingual homographs would be represented in a similar manner as homonyms in a monolingual model: they would consist of a single orthographic pattern that is associated with multiple distinct semantic patterns. Cognates would have a similar representation as polysemous words: one single orthographic pattern that is connected to multiple (highly) overlapping semantic patterns, each of which represents a single sense[8].

## Effects of cross-lingual long-term priming on the processing of cognates and interlingual homographs

The second aim of the experiments presented here was to determine whether recent experience with a cognate or interlingual homograph in one's native language affects subsequent processing of these words in one's second language, as another way to determine how cognates and interlingual homographs are represented in the bilingual lexicon. Experiment 2 indeed demonstrated that there was some evidence for a cross-lingual long-term priming effect. Importantly, the effect of priming was in the predicted direction for both the cognates and the interlingual homographs, although the effects themselves were smaller than expected. For the cognates, priming was facilitative but the difference between primed and unprimed cognates was a non-significant 5 ms. For the

[8]The extent to which these patterns would overlap for each cognate would depend on the extent to which the cognate is used in mostly similar or slightly different contexts in the two languages it belongs to. The cognate "alarm", for example, would consist of highly overlapping patterns that represent the senses of "warning" and "warning system", which are shared in Dutch and English. It would also be associated with a pattern that represents its use in English to mean "alarm clock" and this pattern would overlap less with the other patterns, since in Dutch the word "wekker" would be used to refer to an "alarm clock". This presents another advantage of a distributed connectionist model over a localist model like the BIA+ or Multilink model, as these models assume that cognates have exactly the same meaning across different languages.

interlingual homographs, priming was disruptive, and the difference was a non-significant 10 ms. Crucially, the analyses also revealed that these two effects were significantly different from each other, as predicted, although neither effect was significantly different compared to the 2 ms 'effect' for the baseline condition (i.e., the translation equivalents).

In sum, although cross-lingual long-term priming did not have as strong an effect as expected, Experiment 2 echoed *Poort, Warren & Rodd*'s (*2016*) original findings that a single encounter with a cognate or interlingual homograph in one's native language may influence subsequent processing in one's second language. As discussed previously, this finding provides evidence in favour of the shared-lexicon account: priming in Dutch could only have influenced processing of the cognates and interlingual homographs in English if the two languages are stored in a fully integrated lexicon. On the assumption that long-term priming is caused by a strengthening of the connection between a word's form and its meaning (*Rodd et al., 2013*; *Rodd et al., 2016*), the fact that priming was beneficial for the cognates also suggests that cognates must share both their form and meaning representation. This would fit with the BIA+ model's interpretation of cognates (*Dijkstra et al., 2010*; *Peeters, Dijkstra & Grainger, 2013*), but not the Multilink model (*Dijkstra et al., 2018*), which posits that cognates consist of two language-specific orthographic nodes. Furthermore, since priming was disruptive for the interlingual homographs, it appears that interlingual homographs must also share their form representation. The BIA+ model, in contrast, states that interlingual homographs have separate, language-specific form representations (*Kerkhofs et al., 2006*). If that were true, it seems unlikely that priming would have been observed for the interlingual homographs as there is no reason to assume that strengthening the connection between the Dutch form representation and the Dutch meaning representation would interfere with accessing the English meaning representation through the English form representation.

It seems, then, that these findings would not entirely fit within the localist connectionist frameworks proposed by the BIA+ and Multilink models. Indeed, Rodd et al. (*2013*, *2016*) have suggested that the long-term priming mechanism itself would be most easily accounted for by a distributed connectionist model. Both the facilitative effect of priming for the cognates and the disruptive effect for the interlingual homographs would fit in the distributed framework that we proposed in the previous section, as both cognates and interlingual homographs would consist of a shared form representation in such a model. Accessing the meaning of a cognate in Dutch strengthened the connection between that shared form representation and the shared meaning representation, so priming was beneficial. For the interlingual homographs, in contrast, strengthening the connection between the shared form representation and the Dutch meaning representation at prime proved disruptive when participants then had to access the English meaning representation at test.

It remains unclear exactly why the effect was smaller than predicted, however. It could be the case that priming had a relatively small effect because the 'dominant' native-language meanings of the cognates and interlingual homographs were primed and not the 'subordinate' second-language meanings. Since the participants were more fluent in Dutch and were resident in a Dutch-speaking country at the time of the experiment, the Dutch

interpretations of the cognates and interlingual homographs were most likely dominant for these participants, while the English interpretations were subordinate. Research in the monolingual domain suggests that priming with a single instance of the dominant meaning does not affect subsequent processing of the subordinate meaning (*Betts, 2018*). Therefore, priming the cognates and interlingual homographs in Dutch may not have influenced subsequent access to the English meanings of these items. Other research carried out by *Betts (2018)* suggests that multiple encounters with a word (albeit with its subordinate meaning) increase the long-term priming effect. Perhaps with multiple exposures to the items in Dutch, or if the cognates and interlingual homographs were primed in English, a stronger effect of priming would be observed. Further research would have to examine how the number of exposures at priming and the direction of priming may affect the size of the cross-lingual long-term priming effect.

Finally, priming was predicted to affect the cognates and the interlingual homographs on the assumption that they share their form across languages. However, identically spelled cognates and interlingual homographs are not always also pronounced identically. (Indeed, the mean pronunciation similarity rating for the cognates in these experiments was 5.87 out of 7 and for the interlingual homographs was 5.53.) If participants accessed the phonological form of these words as well as their orthographical form, this slight mismatch in pronunciation in Dutch and English may have interfered with priming. In fact, throughout this manuscript, we have focused exclusively on the effect of orthographic similarity on cognate and interlingual homograph processing in a semantic relatedness task. In the lexical decision literature on cognate and interlingual homograph processing, phonological similarity has been shown to inhibit recognition of target words (*Dijkstra, Grainger & Van Heuven, 1999*; *Lemhöfer & Dijkstra, 2004*). Further research would be necessary to determine what effect phonological similarity has on bilinguals making semantic relatedness judgements.

## CONCLUSION

We set out to examine how bilinguals process cognates and interlingual homographs at a semantic level, using a semantic relatedness judgement task instead of the ubiquitous lexical decision task. The experiments we have presented demonstrate that it is crucial not to forget that cognates and interlingual homographs are in essence just semantically ambiguous words: the bilinguals in our studies appeared to process cognates and interlingual homographs as monolinguals process polysemes and homonyms, respectively. Since processing of such words in the monolingual domain is best modelled using a distributed connectionist approach, we suggest that future endeavours to model the bilingual mental lexicon (re-)explore the viability of such an approach for cognates and interlingual homographs, following in the footsteps of the Distributed Feature Model (*De Groot, 1992*; *De Groot, 1993*; *De Groot, 1995*; *De Groot, Dannenburg & Van Hell, 1994*; *Van Hell, 1998*; *Van Hell & De Groot, 1998*) in the bilingual domain and *Rodd, Gaskell & Marslen-Wilson (2004)* and *Armstrong & Plaut (2008)* in the monolingual domain. In such a model, we suggest cognates would consist of a single orthographic representation and two

highly similar semantic representations within a single broad and stable attractor basin. Interlingual homographs, in contrast, would consist of a single orthographic representation that is connected to two separate and distinct semantic representations that each form deep and narrow attractor basins. Perhaps most importantly, then, our work underscores the argument made by *Degani & Tokowicz (2010)* that bilingual and monolingual researchers should work together in their quest to determine how people resolve semantic ambiguity, both within and between languages.

## ACKNOWLEDGEMENTS

The authors would like to thank René Poort for proofreading the experimental materials, Tessa Obbens and Prof Marc Brysbaert for help with recruitment for Experiment 2 and Dr Rachael Hulme for providing feedback on an early version of this article. We would also like to thank Dr Tamar Degani and Dr Koji Miwa for their helpful comments during peer review.

### Funding

Eva D. Poort was supported by a demonstratorship from the Division of Psychology and Language Sciences, University College London. Jennifer M. Rodd was supported by a grant from the Economic and Social Research Council (No. ES/K013351/1). The funders had no role in study design, data collection and analysis, decision to publish, or preparation of the manuscript.

### Grant Disclosures

The following grant information was disclosed by the authors:
Division of Psychology and Language Sciences, University College London.
Economic and Social Research Council: ES/K013351/1.

### Competing Interests

The authors declare there are no competing interests.

### Author Contributions

- Eva D. Poort conceived and designed the experiments, performed the experiments, analyzed the data, contributed reagents/materials/analysis tools, prepared figures and/or tables, authored or reviewed drafts of the paper, approved the final draft.
- Jennifer M. Rodd conceived and designed the experiments, authored or reviewed drafts of the paper, approved the final draft.

### Human Ethics

The following information was supplied relating to ethical approvals (i.e., approving body and any reference numbers):
The UCL Experimental Psychology Ethics Committee provided approval of the study protocols (Project ID: EP/2017/009).

## Data Availability

For Experiment 1: The pre-registration and all data, scripts and materials are available on the Open Science Framework, http://www.osf.io/ndb7p.

For Experiment 2: The pre-registration and all data, scripts and materials are available on the Open Science Framework, http://www.osf.io/2at49.

## Supplemental Information

Supplemental information for this article can be found online at http://dx.doi.org/10.7717/peerj.6725#supplemental-information.

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
