# Peer review of "Towards a distributed connectionist account of cognates and interlingual homographs: evidence from semantic relatedness tasks"

_PeerJ, doi:10.7717/peerj.6725_

## Round 0.1 · original submission · Minor Revisions

I have received two signed reviews from experts in the field, Tamar Degani and Koji Miwa. Both are incredibly positive about the manuscript. Their feedback should be easily addressed in a revision. I would like to echo one piece of substantive advice from Dr. Degani, who suggested that Experiments 1 & 2 do not contribute to a singular, coherent narrative. Dr. Degani suggested that the experiments should be split into two manuscripts. I leave that choice up to you, but if you do decide to present the experiments in a single manuscript, I hope you will work to improve the coherence of the piece and the transition from Experiment 1 to Experiment 2.

I also want to applaud you on your transparency and openness. Your use of the Open Science Framework was appreciated.

·

Basic reporting

This is a well designed, well conducted and well written study, examining task dependent modulations in cross-lingual effects among proficient adult bilinguals. It further tests for long term cross lingual priming in the same population. The introduction is overall well written, and provides the reader with adequate theoretical and empirical basis to understand the motivation for the current study

When reviewing the literature, the authors mention a single study that utilized the semantic relatedness task. There are at least two other studies that have used this paradigm in the bilingual research (Prior, Degani, Awawdy, Yassin, & Korem, N., 2017, Cognition; Degani, Prior, & Hajajra, 2018, BLC) and both have observed reliable cross-lingual effects (although Interlingual homophones were utilized in these studies). Further, Bowers, Mimouni, and Arguin, 2000, Memory & Cognition) examined long-term priming of cognates. These studies should be incorporated into the text.

Lines 70-74: the transition should be improved. It is not clear why the word type differences imply differential (or special) representations rather than differences in activation patterns (or even response based differences), and this does not seem to be a critical point here. Further, the transition “another important question…” is rather weak.

Paragraph between lines 202-216 – it is interesting to note here that Prior et al. (2017) utilized a cross modal priming, where the cognate/IH was auditorially presented in one language and their non-target language meaning was nonetheless activated and primed the target word. Thus, the use of a Hebrew accent did not preclude the activation of the Arabic meaning of the interlingual homophone.

Experimental design

The two experiments are described in full, with details regarding the participants, the stimuli, and the analysis readily accessible. All relevant materials are available through the OSF website. The interpretation of the findings is grounded, and adheres to the pattern observed in the analysis.

Validity of the findings

The discussion is well organized. The results and their interpretation are highly relevant, and can provide the basis for substantial advancement in modeling of bilingual research.

The motivation for the direct comparisons between cognates and Interlingual homographs is not entirely clear (the 2*2 analyses comparing these two types of words). As the authors note, these comparisons tell us little about the nature of the difference (cognate facilitation vs. IH interference), and the pattern is more clearly captured by the other 2*2 comparisons (cognates vs. control and IH vs. control). Consider removing these comparisons.

In Experiment 2 pairwise comparisons are reported in which the word type effect is examined within the unprimed trials. It is not clear why similar analyses are not reported in addition for the primed trials (unless I have missed this somehow).

Line 623 – Reassuringly… - not clear why this is reassuring?

Line 1399 - the authors interpret their findings as providing a broad replication for the cross lingual long term priming observed in previous research. Given the non significant priming effect in all comparisons, this statement is not entirely clear.

Additional comments

Below I outline a few issues which if addressed I think will improve the paper.

Most critically, although the authors do a remarkable job of leading the reader through the complexities of their experiments, the manuscript is still very long and includes several theoretical issues. I would suggest the authors to consider splitting the manuscript into two, with each focusing on either Experiment 1 or Experiment 2. Although the two experiments used the same materials and criteria, the theoretical questions addressed by each experiment are different. I believe the reader would benefit from a more focused paper, dealing either with task dependency modulations of the effects or with the long-term priming of the words. If the authors decide to keep both together, be sure to clearly mark in the figures which Experiment is being depicted.

Second, although the role of phonology is beyond the scope of the current study, it may nonetheless be valuable to mention it in the discussion section – as overlap in phonology is likely to modulate and contribute to the dynamics of activation of cognates and interlingual homographs.

In general, I think this is a highly interesting paper, bridging the gap between monolingual ambiguity research and bilingual cross-language ambiguity research, and is likely to make substantial theoretical contribution. I look forward to seeing this work published soon.

Tamar Degani

·

Basic reporting

Overall, it was not difficult to follow the story. The manuscript is written in clear and professional English. It is also well structured, with the theoretical story laid out very well. It is respectable that the authors have provided the background with appropriate references including the most recent model of bilingual processing.

Experimental design

The authors conducted relatively rare experiments in bilingual processing research. This is important, as the authors stated, for converging evidence; the bilingual-specific language effects we want to understand should not be task-dependent. It is respectable that the authors have introduced an impressive amount of methodological rigor and that they have also preregistered the study. Overall, because the experiment was designed and described meticulously, replication must be possible with the information provided in the manuscript.

Validity of the findings

The analyses were performed with an up-to-date statistical technique. I can confirm that the data and the corresponding analytical scripts (R codes) are available at the public data repository. It is also clear that the authors applied a decent amount of experimental control and conducted the experiment professionally from the participant recruitment stage till the end.

Additional comments

Summary
This study reports two semantic relatedness experiments with two different focuses: (1) a bilingual vs. monolingual comparison in a semantic relatedness experiment and (2) a long-term priming effect on semantic relatedness performance. In both experiments, inhibitory interlingual homograph effects were observed while the well-known cognate facilitation effects were not observed. The authors conclude that the present results are not in line with predictions of the current localist connectionist models for bilingual processing. Instead, they interpret the results with a distributed connectionist model, motivated by insight from monolingual research: interlingual homographs and cognates are processed in a semantic task just like homonyms and polysemous words respectively.

Evaluation
The unique pattern of results involving both cognates and interlingual homographs will certainly be a valuable addition to our knowledge of bilingual processing. Although I initially felt that the manuscript was slightly long, I soon noticed that it is partially due to the meticulous methodological descriptions, and this is great. Overall, I enjoyed reading this carefully written piece of work, and my impression towards this study is highly positive. I believe that this manuscript will be publishable in PeerJ with only minor revisions. I will be happy if the authors can consider the following minor issues (which are more like my requests) and line-by-line comments.


Minor issues

1. "Trial Number" in the models
Distinguishing lexicon-based processes and task-based processes is a key topic in this study. The authors are perfectly aware that "the semantic relatedness task, like any task, involves decision processes. Consequently, the effects [...] could have been due to response competition" (line 1265-1266). Given this, I am wondering whether the authors have ever included the trial count ("Trial Number" in the data frame?) in the regression models. If not, I am curious how this variable will behave. The motivation is that less automatic task-based effects at the response level must be susceptible to the trial number (e.g., It takes some time for participants to suspect that Dutch knowledge is relevant in the experiment). I believe that more insight can be obtained if the following interactions are tested:

Item type * Trial Number — if the item type effects are lexicon-based, they should be observable from the beginning of the experiment

L1 frequency * Trial Number — If participants consciously start suspecting the involvement of Dutch words, the L1 frequency effect should become larger later in the experiment.


2. Visualization

The violin plots are interesting and certainly informative in that readers can inspect the data distribution and the individual differences across different item types. However, perhaps many readers would like to see the partial effects from the mixed-effects models plotted with confidence intervals. I would like the authors to consider this option in case other reviewers, too, submit the same concern (then the violin plots can be kept in the appendix).


Line-by-line comments

line 136: "The problems with .."
-> I found the word "problems" slightly too strong here. Past lexical decision studies were certainly valuable to the filed; they just had "limitations."

line 142: "a disadvantage for interlingual homographs"
-> I understand what you mean here, but "processing cost" or "prolonged processing" might be less subjective and more appropriate than "disadvantage."

line 340-342: "Experiment 1 was ... relevant to the task."
-> Apparently, the authors are safeguarding against potential effects of the language mode or something similar. It might be better to state the motivation and cite the relevant paper(s) here. For example,

Grosjean, F. (1998). Studying bilinguals: Methodological and conceptual issues. Bilingualism:
Language and Cognition, 1, 131–149

Soares, C., & Grosjean, F. (1984). Bilinguals in a monolingual and a bilingual speech mode:
The effect on lexical access. Memory and Cognition, 12, 380–386.

line 448: "random word generator websites"
-> May I ask what those random word generator websites are?

line 535: "separately"
-> Please remove this repeated word.


Koji Miwa

---

## Round 0.2 · accepted · Accept

I thank you for your thoughtful responses to the reviewer critiques. I trust that this paper will spur much more research on the complex interactions within the bilingual lexicon.

#